# Electric-field-assisted proton coupling enhanced oxygen evolution reaction

Xuelei Pan [1,2], Mengyu Yan[1] ✉, Qian Liu[3], Xunbiao Zhou[1], Xiaobin Liao [1], Congli Sun[1], Jiexin Zhu[1], Callum McAleese[4], Pierre Couture [4], Matthew K. Sharpe [4], Richard Smith[4], Nianhua Peng[4], Jonathan England[4], Shik Chi Edman Tsang [2] ✉, Yunlong Zhao [5,6] ✉ & Liqiang Mai [1] ✉

The discovery of Mn-Ca complex in photosystem II stimulates research of manganese-based catalysts for oxygen evolution reaction (OER). However, conventional chemical strategies face challenges in regulating the four electron-proton processes of OER. Herein, we investigate alpha-manganese dioxide ($\alpha$-MnO$_2$) with typical Mn$^{IV}$-O-Mn$^{III}$-H$_x$O motifs as a model for adjusting proton coupling. We reveal that pre-equilibrium proton-coupled redox transition provides an adjustable energy profile for OER, paving the way for in-situ enhancing proton coupling through a new "reagent"− external electric field. Based on the $\alpha$-MnO$_2$ single-nanowire device, gate voltage induces a 4-fold increase in OER current density at 1.7 V versus reversible hydrogen electrode. Moreover, the proof-of-principle external electric field-assisted flow cell for water splitting demonstrates a 34% increase in current density and a 44.7 mW/cm² increase in net output power. These findings indicate an in-depth understanding of the role of proton-incorporated redox transition and develop practical approach for high-efficiency electrocatalysis.

Oxygen evolution reaction (OER) is one of the most common reactions for molecular oxygen formation. Owing to its high energy barrier and sluggish reaction kinetics, OER becomes the rate-determining step in many biologic and chemical processes[1–3], including artificial photosynthesis[4–7], electrocatalytic water splitting[8,9], rechargeable metal-air battery[10,11], etc. Understanding the reaction processes and developing highly efficient OER catalysts are the key goals for advancing the aforementioned processes[12,13]. Inspired by the highly efficient enzyme catalysis, many efforts have been made to synthesise catalysts mimicking the enzyme structures and functions[14,15]. Taking artificial photosynthesis as an example, although the OER pathways in artificial photosynthesis are similar to the natural process, the efficiency via synthetic catalysts is lower than that via Photosystem II (ref. 6), an enzyme that extracts electrons from H$_2$O and feeds them into an electron-transport chain for the chemical synthesis using the input of solar energy[16]. It is widely believed that this efficiency gap lies in the structural differences between synthetic catalysts and the core of enzymes[16,17]. Therefore, designing and synthesising artificial catalysts that mimic the core structure of enzymes or analogues becomes a promising direction[17,18]. Researchers found that the oxygen evolution proceeds specifically at the catalytic centre of the Mn$_4$CaO$_x$ cluster in PSII (Fig. 1a)[7,19]. It facilitates synthesising a series of oxygen-evolving complexes mimicking the Mn$_4$CaO$_x$ structure[18]. For instance, [Mn$_{12}$O$_{12}$(O$_2$CMe)$_{16}$(H$_2$O)$_4$] was synthesised as a stable homogeneous water oxidation electrocatalyst operating at pH 6 with an exceptionally low overpotential of 334 mV[20]. Besides, some manganese oxides (e.g. $\alpha$-MnO$_2$)[21,22] have also been considered promising biomimetic catalysts for OER due to their low cost, facile synthesis and structural similarity

[1]State Key Laboratory of Advanced Technology for Materials Synthesis and Processing, International School of Materials Science and Engineering, Wuhan University of Technology, Wuhan 430070, P.R. China. [2]Wolfson Catalysis Centre, Department of Chemistry, University of Oxford, Oxford OX1 3QR, UK. [3]School of Materials Science and Engineering, Zhejiang University, Hangzhou 310027, P. R. China. [4]UK National Ion Beam Centre, University of Surrey, Guildford, Surrey GU2 7XH, UK. [5]Dyson School of Design Engineering, Imperial College London, London SW7 2BX, UK. [6]National Physical Laboratory, Teddington, Middlesex TW11 0LW, UK. ✉e-mail: ymy@whut.edu.cn; edman.tsang@chem.ox.ac.uk; yunlong.zhao@imperial.ac.uk; mlq518@whut.edu.cn

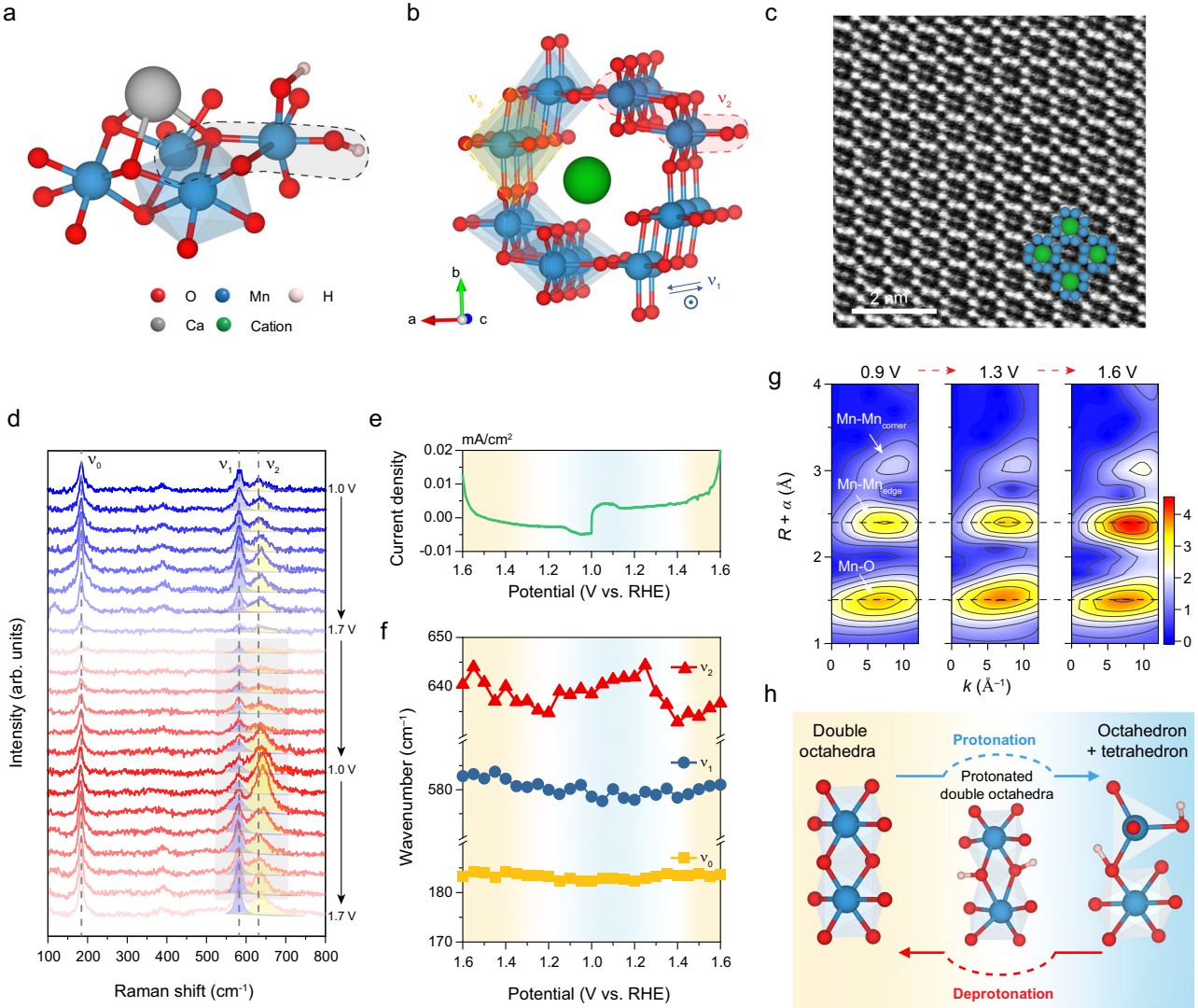

**Fig. 1 | Atomic structure of α-MnO₂ and in situ monitoring of the structural evolution of α-MnO₂ in the OER process. a, b** Ball-and-stick models of Mn₄CaOₓ complex and cation-accommodated α-MnO₂. The Mn$^{IV}$-O-Mn$^{III}$-H$_x$O motif and a similar di-μ-oxo di-manganese structure are highlighted in grey and red dashed bands, respectively. The atomic structures and vibration directions corresponding to three typical Raman peaks v₀, v₁, and v₂ are also labelled in (**b**). **c** HAADF image of cation-accommodated α-MnO₂ along the [001] zone axis, revealing the atomic structure of 2 × 2 tunnels surrounded by eight Mn atomic columns (blue spheres) and with accommodated cations (green spheres) in the tunnels. **d** In situ Raman spectra of α-MnO₂ electrode recorded in 1 M KOH with a potential window of 1.0–1.7 V vs. RHE and a potential interval of 0.1 V. The voltage and arrows on the right represent the potential (vs. RHE) and scan direction of CV measurement. Raman bands with three main contributions at -183 cm⁻¹, -580 cm⁻¹ and -635 cm⁻¹ are marked as v₀, v₁, and v₂, respectively. The multi-peak deconvolution of Raman spectra is shown in the selected potential range of 1.6 – 1.0 – 1.6 V vs. RHE indicated by a grey background. **e** Simultaneous voltammogram in the potential range. **f** The peak shift of v₀, v₁ and v₂ in the selected potential range. **g** In situ Mn K-edge EXAFS WTs during OER process 0.9, 1.3 and 1.6 V vs. RHE. The colour bar is WT moduli. The dashed line is drawn to illustrate the evolution. **h** Predicted schematics of the evolution of the di-μ-oxo di-manganese structure in the α-MnO₂ during the OER process. The schematic of the structure motif changes in a proton-coupled redox transition. The protonation is the adsorption of protons on the bridging oxygen, the following transformation represents the degradation from octahedral to the tetrahedral ligand, and then the deprotonation corresponds to the structural recovery in the anodic process.

to the Mn$^{IV}$-O-Mn$^{III}$-H$_x$O motif (Fig. 1a, highlighted in grey dashed band) – the functional core in Mn₄CaOₓ cluster for oxygen evolution[23–25], although their structural and functional similarity and the corresponding biomimetic strategies have not been thoroughly studied.

Although significant efforts have been made to mimic the core structure of enzymes, the catalytic efficiency of the synthetic catalysts is still far lower than that of enzymes[16]. This phenomenon inspired us to investigate other underlying factors that may affect the catalytic activity beyond structures. We first review the catalytic process in organisms from the perspective of molecular dynamics. The proton-coupled electron transfer (PCET) is a redox process that involves the simultaneous transfer of electrons and protons in an enzymatic

catalytic reaction, which leads to more favourable energetics compared to the sequential pathways (decoupled proton-electron transfer)[26]. Due to the PCET, a kinetically rapid redox transition can occur in the absence of a significant overpotential, ensuring that the activity of enzymes is sufficient to maintain a high and constant metabolism in the organism under mild conditions[27,28]. However, for synthetic catalysts, especially the transition metal oxides, the OER usually follows the sequential pathway due to the high proton affinity of the electrode surface: they can hold the local protons or other cations through adsorption or noncovalent interactions from the electrolyte by altering the oxidation state of the metal centre[29,30]. Therefore, enhancement of deprotonation may favour the PCET

pathway for these catalysts[31,32]. Research into proton insertion/extraction within the $MnO_2$ lattice has been ongoing for many years, particularly since the advent of alkaline batteries[33]. However, a more profound comprehension of the intricate and diverse properties of protons involved in various electrochemical systems, as well as the proton-electron processes, is imperative to attain enhanced control over the thermodynamics of proton reactions. In addition to the PCET, the local field in the biological system is another 'smart reagent' to promote the reaction[34,35]: Researchers have recently discovered that the endogenous electric fields exist widely in microorganisms, plants, and animal cells, influencing the biological activities and cellular behaviour of tissues and organs[36,37]. They can affect many biological processes and tune the kinetics of enzyme catalysis, such as the charge transfer in photosynthesis[38,39]. These discoveries may offer us new routes to further optimise or develop novel OER catalysts.

Herein, we take the cation-accommodated $\alpha$-$MnO_2$ nanowires as a prototype of a synthetic catalyst, study the structural and functional role of μ-oxo di-manganese structure and protons played in the oxygen evolution process, and report a facile electric-field-assisted strategy for advancing OER. We investigate how the proton and redox transition tailor the alkaline OER by tracing the proton-electron transfer process using in situ Raman spectra and elastic recoil detection (ERD) hydrogen spectra. Inspired by the endogenous electric field from the biological systems, a modulated external electric field is applied to enhance the deprotonation and proton coupling for high-efficiency electrocatalysis, featuring an exceptionally low overpotential of 360 mV (at 100 $mA/cm^2$), which is significantly superior to that without electric-field (current density at 1.7 V vs. reversible hydrogen electrode (RHE) increase from 141 to 704 $mA/cm^2$) and is comparable to noble metal oxide (e.g. $IrO_2$[40]) at similar conditions. Moreover, based on a new field-enhanced flow cell system, this strategy is demonstrated to be effective in a centimetre-sized electrolyser in which the electric field increased the current density of overall water splitting by 34% using the $MnO_2$ film electrode.

## Results and discussion
### The proton coupling of di-μ-oxo in $\alpha$-$MnO_2$

The typical structural feature of $\alpha$-$MnO_2$ is the 2 × 2 tunnel structure stabilised by cations (Fig. 1b). Structure characterisations and element analysis of $\alpha$-$MnO_2$ are presented in Supplementary Fig. 1a, b. The high-angle annular dark-field (HAADF) image shows the 2 × 2 tunnels surrounded by eight Mn atomic columns and cations in the tunnels (Fig. 1c). According to the XPS analysis (Supplementary Fig. 1c, d) and elemental mapping (Supplementary Fig. 2), the accommodated cations include $K^+$ ions. Besides $K^+$ ions, it is acknowledged that other cations such as protons ($H^+$) or hydroniums ($H_3O^+$) could also be accommodated in the 2 × 2 tunnels in hydrothermally synthesised $\alpha$-$MnO_2$ (ref. 20,41,42). These accommodated cations not only stabilise tunnel structures and balance the charges but also favour the formation of mixed-valence states ($Mn^{3+}/Mn^{4+}$)[23], rendering the di-μ-oxo di-manganese (Mn-(μ-oxo)$_2$-Mn) in $\alpha$-$MnO_2$ more similar to the $Mn^{IV}$-O-$Mn^{III}$-$H_xO$ in $Mn_4CaO_x$ complex (Fig. 1a, b). With these characterisation results, we first investigate the roles of this similar structure in $\alpha$-$MnO_2$ throughout the oxygen evolution process. The OER performance of $\alpha$-$MnO_2$ nanowire thin film was measured in 1 M KOH electrolyte using a conventional three-electrode configuration. The polarisation curve shows a moderate overpotential of 480 mV and a Tafel slope of 95 mV/dec (Supplementary Fig. 3), consistent with the data reported in the literature[23].

To investigate the structural evolution, we recorded the Raman spectra during the cyclic voltammetry (CV) measurement within a potential range from 1.0 to 1.7 V vs. RHE (Fig. 1d and Supplementary Fig. 4). By adjusting the CV scan rate, each spectrum was measured over a 50 mV potential range. This approach provided sufficient resolution to discern the sequential reaction processes and observe

structural evolution associated with the electrochemical reactions. Two cycles were measured to provide a comprehensive understanding of structural evolution, taking into account the initial states before electrochemical conditioning, thereby ensuring robust and convincing results. As shown in Fig. 1d, three typical Raman peaks at ~183, ~580 and ~635 $cm^{-1}$ are observed and marked as $v_0$, $v_1$, and $v_2$, respectively, which correspond to vibration modes marked in the Fig. 1b. To quantify each peak, Raman signals at ~580 and ~635 $cm^{-1}$ are deconvoluted and fitted. When the potential reaches ~1.6 V, we notice large oxygen bubbles generating from the electrode surface (a typical oxygen evolution reaction feature), scattering the laser and causing a weak Raman signal. While the spectrum intensity is low, it provides evidence of the presence of oxygen bubbles and offers insight into the structural characteristics. Therefore, we select the potential range 1.6 − 1.0 − 1.6 V to study the reversible process by tracking the current density changes (Fig. 1e) and the Raman peak shifts (Fig. 1f). As shown in Fig. 1e, the weak reversible redox peaks (~1.15 − 1.0 V) are observed (marked by blue bands), corresponding to the valence transition of $Mn^{3+}/Mn^{4+}$ (ref. 23), followed by the OER after 1.5 V. In the selected potential range, $v_0$ and $v_1$ remain at almost the same wavenumbers while the wavenumber of $v_2$ shows a fluctuation. As shown in Fig. 1f, this process can be divided into three continuous parts: 1. Cathodic process (1.6 − 1.2 V); 2. Redox process (1.2 − 1.0 − 1.4 V); 3. Anodic process (1.4 − 1.6 V). In process 1, $v_2$ gradually decreases from 640 to 635 $cm^{-1}$. Then in process 2, $v_2$ sharply increases to ~640 $cm^{-1}$ and keeps at this level until 1.2 V during this process. This process ends with a decrease of $v_2$ to 633 $cm^{-1}$ at 1.4 V. In the following process 3, $v_2$ shifts back to 637 $cm^{-1}$. It is noteworthy that the $v_2$ peaks exhibit low reversibility in terms of wavenumber, attributed to the irreversible structure transformation induced by potential Jahn−Teller distortion. Following the second anodic process (process 3), the profiles, including peak position and intensity, revert to the typical doublet vibration bands characteristic of $\alpha$-$MnO_2$. In addition to the variation of $v_2$, the peak at ~385 $cm^{-1}$ also reappears and shows obvious intensification after the electrode potential increases to 1.4 V. This typical vibration band can also be found in MnOOH and is attributed to the Mn-OH features[43], which indicates the Mn-O bending is enhanced in the anodic process.

The shift to high wavenumber and intensity increase of $v_2$ band indicates the transition from octahedral structure to tetrahedral structure (similar to the spinel structure of $Mn_3O_4$)[44], which is attributed to the Jahn−Teller effect of $Mn^{3+}$ induces the distortion from the octahedral to the tetrahedral ligand at low potential (1.2 − 1.0 V). In alkaline electrolytes, the electron-proton mechanism[45] was used to describe the redox process of $Mn^{3+}/Mn^{4+}$, where protons are introduced to the lattice and coupled with the bridging oxygen to form O−H bonds in the cathodic process. Hence, the protons are supposed to be incorporated into this structure evolution process. In the subsequent anodic process (1.2 − 1.4 V), the local tetrahedral structure [$MnO_4$] transforms back to octahedral [$MnO_6$] accompanied by the proton extraction, and $v_2$ reversibly shifts to the lower wavenumber and the intensity decreases to the original value. The $v_0$ and $v_1$ maintain their position during the transition process, indicating that the tunnel structures are stable, and this transition process is supposed to occur in local regions such as at the superficial surface of electrode. Herein, in situ Raman spectra demonstrate the evolution of [$MnO_6$] with the proton adsorption and extraction, i.e., the valence transition of Mn.

To further analyse the structural changes in the coordination environment of Mn atoms, in situ X-ray absorption spectroscopy (XAS) (Supplementary Fig. 5) was performed during the anodic process at three potentials (0.9, 1.3, and 1.6 V). In order to attain high-quality spectra, time resolution was not prioritised for the in situ XAS measurements. Instead, multi-potential measurements were employed. The wavelet transform (WT) of extended X-ray absorption fine structure (EXAFS) spectra are shown in Fig. 1g. In the contour image, the maximum intensity regions at ~1.5 and ~2.5 Å correspond to Mn-O bond

and the Mn-Mn coordination in edge-shared [MnO$_6$] octahedra (di-μ-oxo bridged), respectively (Fig. 1g). Compare the spectra at 0.9 and 1.6 V, Mn-O bonds and Mn-Mn coordination shift to low apparent radial distance, indicating a decrease in Mn-O length and the Mn-Mn distance. At 1.3 V, a slight increase in Mn-O distance is observed, which could be linked to the elongation of z ligand bonds resulting from Jahn-Teller octahedral distortion. The enhanced scattering intensity of edge-shared Mn-Mn coordination demonstrates the recovery of [MnO$_6$] octahedra. This suggests that the proton extraction is embodied in the evolution of edge-shared [MnO$_6$], which is consistent with the recovery process of ν$_2$ in Raman results. Combined with in situ Raman results, di-μ-oxo between [MnO$_6$] octahedra is supposed to be the main sites for the pre-catalysis process (including the deprotonation process). Therefore, in situ spectra results mainly demonstrate that the incorporated protons couple with the structure evolution of edge-shared [MnO$_6$] octahedra accompanied by redox transition of Mn. We used density functional theory (DFT) calculations to find the adsorption sites of protons and built different models of proton adsorption on di-μ-oxo and mono-μ-oxo sites (Supplementary Figs. 6 and 7). We found that the proton adsorption on di-μ-oxo is thermodynamics spontaneous, while adsorption on mono-μ-oxo site is an energy unfavourable situation, indicating that di-μ-oxo acts as Brønsted basic site. It is also interesting to find the break of di-μ-oxo bridges with protonated terminal oxygen sites, resulting in low-coordinated corner-shared [MnO$_x$] polyhedra (Supplementary Fig. 8). It can be recovered with the deprotonation of terminal oxygen, and break again with *OOH formation by nucleophilic attack. Such results on proton adsorption sites and the related structural evolution effectively demonstrate the phenomenon observed by in situ spectroscopy characterisations. Based on the analysis above, the structure evolution of [MnO$_6$] octahedra in the redox circulation is presented in Fig. 1h. In detail, during the cathodic process, the introduced protons couple with the bridging oxygen in edge-shared double [MnO$_6$] octahedra, followed by the transformation from octahedral to the partial tetrahedral ligand. In the anodic process before OER, the bridging oxygen deprotonates, and the structure recovers from tetrahedron-octahedron to the bridged double octahedra.

To quantitatively investigate the intercalation/extraction process of protons, we conducted elastic recoil detection (ERD) and Rutherford backscattering spectrometry (RBS) analysis with a 2 MeV $^4$He ion beam on the different samples (Fig. 2a–c, Supplementary Fig. 9 and Supplementary Table 2). We compared two set samples conditioned at 0.9 and 1.6 V vs. RHE to ensure that completely protonated and deprotonated states were measured. The RBS spectrum of the pristine sample show that α-MnO$_2$ composition fits the spectrum well. The slight disagreement at the Mn edge (1.5 MeV) is due to the randomly distributed nanowires morphology. An edge at 1.3 MeV in the spectra indicates the presence of about 3% K in the samples, demonstrating the cations in the tunnel structure. The ERD spectrum suggests H concentrations of 5.5% at the very surface, around 800 TFU, and 7% beneath this in the lattice (Fig. 2d–f). This suggests that the pristine MnO$_2$, despite being synthesised in an aqueous environment, is not heavily protonated. This observation explains why the first anodic process during in situ Raman measurement did not exhibit significant structural evolution (Fig. 1d). The RBS and ERD analysis of the other samples show that the sample treated at 0.9 V has the highest H concentration (13.5 – 12%), demonstrating the intercalation of protons in the low potential region. The sample treated at 1.6 V has a lower H concentration (6.0 – 7.5%) following deprotonation at a high anodic potential. Due to these samples being treated in the same electrolyte for the same time, the apparent difference in H concentration also eliminated the potential signal from lattice water. The consistent concentration of K$^+$ effectively rules out the possibility of K$^+$ intercalation or adsorption from the electrolyte. Herein, we prove that protons are extracted at a high potential state ( > 1.2 V), while at a low

potential state ( < 1.1 V), protons are intercalated into the lattice, resulted in a doubled lattice proton concentration.

In addition to the amount of proton in MnO$_2$ lattice, the reversibility of proton cycling is also a remained issue. To uncover this issue, a sequential X-ray absorption near edge structure (XANES) and EXAFS spectra of MnO$_2$ at 0.9 and 1.6 V vs. RHE were recorded. The XANES spectra show weak peaks at pre-edge (~6542 eV) and a strong absorption edge peak at ~6562 eV (Fig. 2g). The weak peaks are attributed to the electric dipole forbidden transition from 1$s$ to unoccupied 3$d$ orbital arising from the non-centrosymmetric [MnO$_6$] octahedra, and the strong peak is the dipole-allowed transition from 1$s$ to 4$p$ orbital. Accordingly, the transition of 1$s$ electron to $t_{2g}$ and $e_g$ orbitals results in the split peaks in the pre-edge region, denoted by P$_1$ and P$_2$, respectively. When increasing potential to 1.6 V, P$_2$ peak shows an obvious increase with the absorption edge shift to higher energy, attributed to the unoccupied $e_g$ orbital in a high oxidation state (the bound state of Mn$^{4+}$ is $t_{2g}^3 e_g^0$). It is worth noting the absorption edge shift is quasi-reversible when switching between high and low potentials, demonstrating a partial redox transition of Mn$^{3+/4+}$. The Fourier transforms of Mn K-edge $k^3\chi(k)$ spectra are shown in Fig. 2b. Three strong Fourier transform peaks are observed, one located at ~1.5 Å represents the Mn-O bonds in [MnO$_6$] octahedra and the other two peaks around ~ 2.4 and 3.0 Å correspond to the edge-shared (di-μ-oxo-bridged) and corner-shared (mono-μ-oxo-bridged) Mn-Mn shells[46], respectively. According to the fitting results (Fig. 2i and Supplementary Fig. 10, Supplementary Table 2), for the samples at 0.9 V, and the length of edge-shared Mn-Mn becomes longer, which can be attributed to the strong interaction between bridging oxygen and protons and the structure is then distorted. In contrast, corner-shared Mn-Mn shows irregular variation. To the sequence details, it can be found that Mn-Mn$_{edge}$ shows more obvious reversible change with potential switching, further proving the coupling of proton to di-μ-oxo. It also points to a possible issue that the irregular structure variation could limit the proton cycling. Two obvious phenomena, smaller P$_2$ peaks and unchanged Mn-O coordination, could also lead to this conclusion. To address this issue, an in-depth analysis of the proton coupled redox transition in electrochemical process should be performed.

## Proton-electron transfer adjusted by external electric field

After identifying the proton-incorporated redox transition and structure evolution process (i.e., pre-catalysis process before oxygen evolution), we then discuss the oxygen evolution process. As the analogue of the oxygen-evolving complex in Photosystem II, the OER pathway on α-MnO$_2$ is proposed as an adsorbate evolution mechanism, including three critical intermediates *OH, *O, *OOH. The lattice oxygen mechanism was ruled out due to no feature of elevated O 2$p$ band and the high free energy change of direct coupling of two oxygen sites[47] (Supplementary Fig. 11). It is worth noting that we found the dual sites absorption state of *OOH (Supplementary Figs. 12,13), attributed to Langmuir–Hinshelwood (LH) mechanism[48]. LH mechanism on MnO$_2$ resembles Mn$_4$CaO$_x$ in PSII, demonstrating a structural similarity induced by similar absorbates configuration. In the adsorbate evolution mechanism, OH$^-$ is the oxygen source and the proton acceptor. The deprotonation process is known to be important for OER, and if the deprotonation of *OH or *OOH is limited, it will directly affect the potential-determining step (PDS) (see details in Supplementary Notes). We calculated and compared Gibbs free energy change of four steps, ΔG$_1$, ΔG$_2$, ΔG$_3$, and ΔG$_4$ with different proton configurations (Supplementary Fig. 14). Considering various proton configurations on the surface and internal structures, we aimed to provide a comprehensive evaluation of how proton transfer influences oxygen evolution. ΔG$_3$ reveals an intriguing phenomenon in the adsorption structure of *OOH. In models without surface protons, *OOH is found to be absorbed by two adjacent Mn atoms (Supplementary Fig. 13), resulting in smaller ΔG$_3$ values compared to other configurations. This

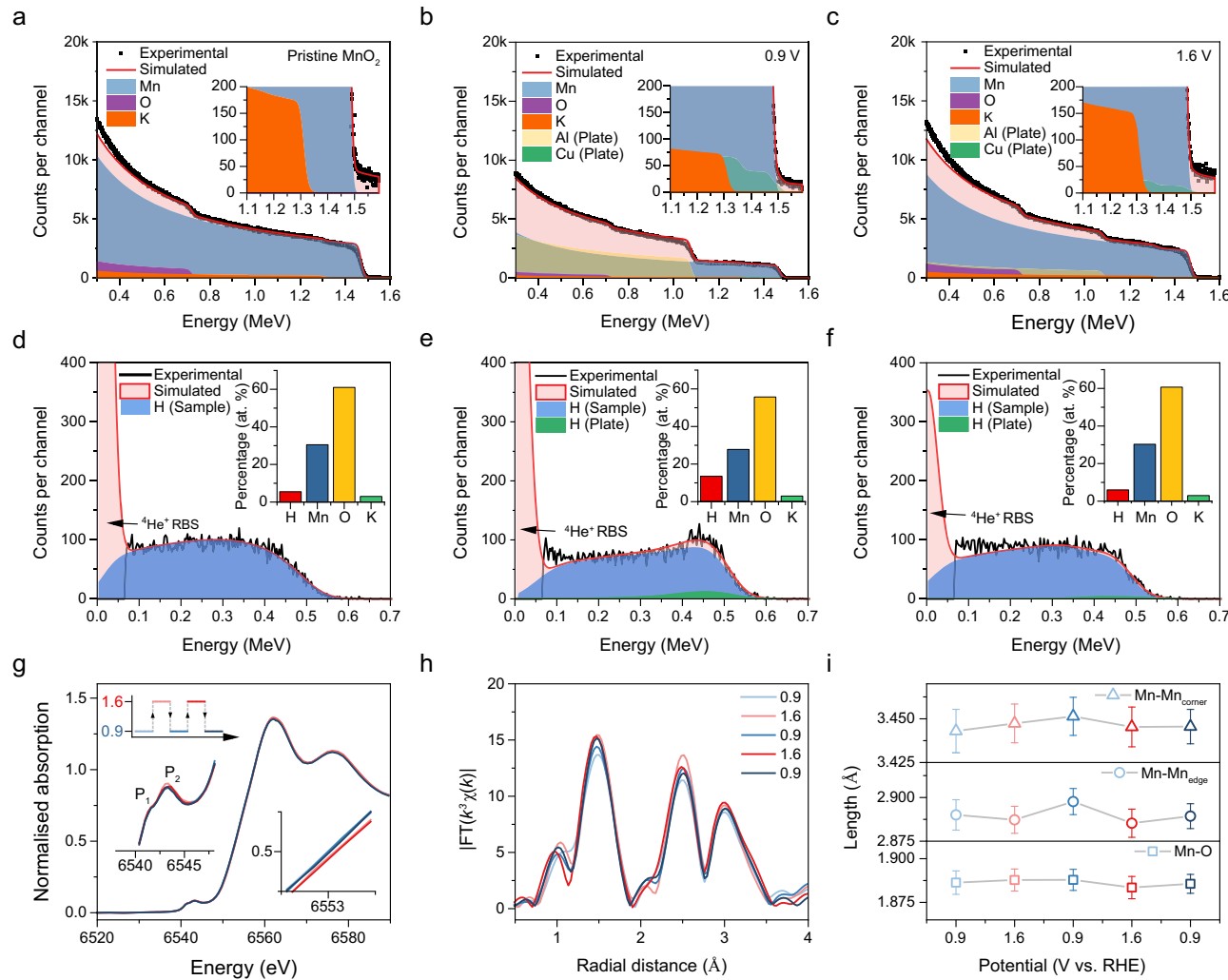

**Fig. 2 | The characterisation of proton incorporated in OER process of α-MnO₂.**
**a–c** The Rutherford backscattering spectrometry spectra for the pristine MnO₂ and MnO₂ samples treated by chronoamperometry at 0.9 and 1.6 V vs. RHE. **d–f** The corresponding ERD hydrogen spectra. Inset: the percentage of different elements present on the surface of the MnO₂. **g** Mn K-edge XANES spectra of MnO₂ measured by multi-step chronoamperometry at 0.9 and 1.6 V vs. RHE. The upper left inset is the illustration of multi-step potential measurement, and the colour corresponds to the curves. The lower left inset is the pre-edge region with a doublet. The lower right inset shows the shift of absorption at half-normalised intensity. **h** The corresponding Fourier transforms of Mn K-edge EXAFS spectra using $k^3$ weighting. **i** The plot of bond/shell length fitted from EXAFS spectra. Mn-Mn$_{edge}$ and Mn-Mn$_{corner}$ represent edge-shared Mn-Mn coordination and corner-shared Mn-Mn coordination respectively.

interesting finding also occurs in models with no tunnel protons but protons on opposite Mn sites. This suggests that the deprotonated surface favours the addition of OH⁻ to *OH. Conversely, if the oxygen on the target Mn sites is protonated, the formation of *OOH, as well as subsequent oxygen molecule formation, is difficult, leading to an overpotential of ~0.8 V (Supplementary Fig. 15). Additionally, tunnel protons can adjust the overpotential by affecting the adsorption free energy, although they cannot modify adsorbates (Supplementary Fig. 16). In conclusion, we find that deprotonated surface states are crucial for forming dual-site O-O and achieving a moderate theoretical overpotential. If considering the intermediates on α-MnO₂ surface as a motif in the whole structure, the deprotonation process is thus determined by both the redox properties of α-MnO₂ surface and H⁺/OH⁻ concentration. This process can also be described as that α-MnO₂ can be further deprotonated if the pH of the electrolyte on the interface is higher than the pKa of the attached proton. The deprotonation of α-MnO₂ not only provides the active sites for adsorption but also adjusts the energy profile for the following OER.

To understand the PCET process of OER, the OER activity of MnO₂ at different pH was measured (Supplementary Fig. 17a, b). Reaction order ($\rho$) can be determined by the linear relationship between log $j$ and pH, $\rho = \left(\frac{\partial \log j}{\partial \mathrm{pH}}\right)_E = -\left(\frac{\partial E}{\partial \mathrm{pH}}\right)_i / \left(\frac{\partial E}{\partial \log j}\right)_{\mathrm{pH}}$, where $j$ is current density, E is potential versus RHE. The fitted slope value is 1.03 ± 0.06, indicating the reverse first-order dependence on H⁺. Hence, the rate-determine step of α-MnO₂ for OER is decided by the concentration of H⁺ involved in reaction, resulting in a strong pH dependent OER activity (i.e., decoupled proton-electron transfer). CV curves at different pH also provide some information about redox transition of Mn (Supplementary Fig. 17c, d). The separation of oxidation and reduction peak potential shows a dependence on pH and scan rate. With the increase of pH, the width of CV peaks and the redox potential separation decreases at a specific scan rate, demonstrating a decreased polarisation of proton-electron reaction at the resting state before OER. The results above demonstrate the OER process on the MnO₂ surface is an decoupled proton-electron transfer reaction which shows inverse first-order dependence on H⁺ concentration. This also indicates that increasing the H⁺ acceptor can adjust the electron and proton coupling to enhance OER. Herein, the energy profile of the OER process is dominated by the proton configuration, and maintaining a circulation

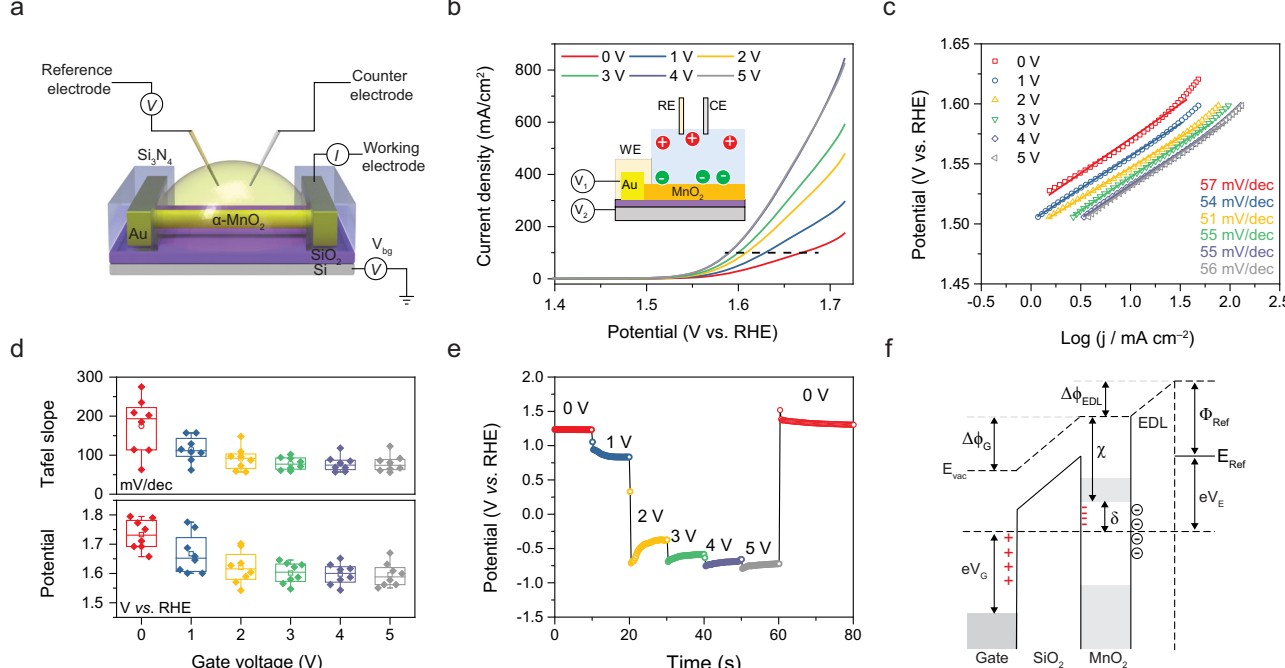

**Fig. 3 | The electrochemical performance of the single α-MnO₂ nanowire device.**
**a** The schematic diagram of the single nanowire electrocatalytic device where a single α-MnO₂ nanowire is connected to the Au microelectrodes with Si₃N₄ as the insulating layer. **b** Polarisation curves and (**c**), Tafel plots of the single α-MnO₂ nanowire at different gate voltages. Inset: the schematic illustration of the working principle of gate voltage. V₁ represents the potential applied to the working electrode and V₂ is the gate voltage. **d** The statistics results of onset potential and Tafel slope plots at different gate voltages from different devices. The error bars represent the standard errors. **e** The gate voltage-tuned open circuit potential of the single α-MnO₂ nanowire. **f** Energy diagrams of MnO₂ electrochemical system with applied positive back gate voltage. The symbols in the diagram are vacuum level ($E_{vac}$), work functions of the back gate ($\Phi_G$) and reference electrode ($\Phi_{ref}$), electron affinity of MnO₂ ($\chi$), electrical double layer (EDL), and vacuum level shifts in SiO₂ ($\Delta\phi_G$).

of lattice and surface protons will contribute to moderate adsorption energy for OER thermodynamics.

Based on the above analysis, regulating the proton coupling could be considered as a more proactive strategy to affect the OER performance. To figure out how to tune the proton-incorporated redox transition, we first review the biological system: In natural photosystems, the membrane potential induced by the ion concentration difference and protein architecture acts as a bioelectric field (endogenous electric field). This field effect is demonstrated to accelerate mass transfer as well as reaction dynamics[49–52]. The electric field-induced enzyme ketosteroid isomerase was demonstrated to accelerate the enzyme catalysis, which can result in a 10⁵-fold enhanced reaction rate[53]. Meanwhile, an external electric field was also proposed as a smart reagent to boost chemical reactions[54] due to its ability to regulate the ionicity of chemical bonds. Inspired by this, we proposed an idea to adjust the proton acceptor (OH⁻) concentration and tune the proton-electron mechanism by applying an external back-gate electric field.

To test the aforementioned idea, a single α-MnO₂ nanowire system was built to precisely measure the electric field modulation. We fabricated single α-MnO₂ nanowire electrocatalytic devices on Si/SiO₂ wafers, constructing a "field-effect transistor"-like configuration[55,56] (Fig. 3a). The gate electrode was connected to the heavily doped silicon substrate with the silicon oxide (300 nm) as an insulating layer. The morphology of single α-MnO₂ nanowire device and the measurement configuration are shown in Supplementary Figs. 18–20. We applied a positive external electric field to α-MnO₂ nanowire in 1 M KOH aqueous solution during an electrochemical measurement. As shown in Fig. 3b, we tested the anodic polarisation curves of individual α-MnO₂ nanowire under different gate voltages. The pristine α-MnO₂ displays a moderate overpotential, 440 mV at 100 mA/cm². Herein, to make the results from a single α-MnO₂ nanowire comparable to the

results from macro-sized measurement, the overpotential at a current density of 100 mA/cm² was chosen as the evaluation criteria. The overpotential of a single α-MnO₂ nanowire goes through a dramatic decrease after applying a positive gate voltage. When increasing the gate voltage to 5 V, the overpotential decreases to 360 mV. This performance reaches a superior level among the manganese oxides-based OER catalysts[23]. Meanwhile, the Tafel slope remains stable at around 55 mV/dec (Fig. 3c), indicating the unchanged reaction path and mechanism. The gate leakage current was recorded at a constant gate voltage, which remains below 1 nA when applied 5 V gate voltage (Supplementary Fig. 21). These results demonstrate a robust stimulating effect of the external electric field. The statistical results also prove that overpotential and Tafel slope decrease dramatically when the gate voltage is below 3 V and approach equilibrium at a higher gate voltage (Fig. 3d). In addition to the overpotential and Tafel slope, the open-circuit potential also varies with gate voltage (Fig. 3e) with a similar equilibrium occurring when gate voltage reaches 3 V.

The shift of open-circuit potential can be attributed to the concentrated anions (OH⁻) at the surface of the nanowire, and the injected electrons adjust the Fermi level[57]. Hence, the equilibrium is considered to be saturated with the concentrated proton acceptor. Figure 3f shows diagram illustration of energy level with positive gate voltage. When applied a positive gate voltage, the energy level shifts due to the charge accumulation at the interface, following Poisson's equation. In our case, a large gate electrode works on both channel material (MnO₂ nanowire) and electrolyte. The back gate energy level shift ($\Delta\phi_G$) is charging through polarisation of the insulating layer (SiO₂) and the electrical double layer shift ($\Delta\phi_{EDL}$) is through anion (OH⁻) accumulation on the electrode surface charging the double layer. Based on the energy level diagram, Fermi level shift can be estimated by the difference of relative offset to conduction band bottom, $\Delta\delta = \delta_0 - \delta = e(V_E - V_{E0})$. The overall shift can be calculated by the

charge coupling with back gate and electrical double layer[58]. The total charge $Q_w$ is expressed as $Q_W = C_G V_G + C_{EDL} V_E$, which indicates a linear relationship between $V_E$ and $V_G$, $\frac{\partial V_E}{\partial V_G} = -\frac{C_G}{C_{EDL}}$, where $C_G$ and $C_{EDL}$ represent gate and EDL capacitance. Supplementary Fig. 22 shows the linear relationship between gate voltage and electrode potential and the slope is fitted to be −0.6. Consequently, we derived a numeric expression of energy shift induced by gate voltage $\Delta\delta = 0.6eV_G$.

Considering the first-order pH dependence for pristine α-MnO$_2$, the dependence of gate voltage-induced overpotential reduction is an analogy. According to the linear relationship between gate voltage and overpotential, we attribute the similar rate constant to the pH dependence, which indicates that the deprotonation process is still limited. Meanwhile, when the gate voltage increases to 3 V, the overpotential shows approximately the "zero-order" dependence of gate voltage accompanied by the unchanged Tafel slope due to the highly concentrated proton acceptor. This phenomenon indicates that by applying a high gate voltage, the deprotonation process is no longer restricted, and the electrons and protons transfer is coordinated and cannot be further regulated (Supplementary Fig. 23). This means that the extraction process of protons is enhanced, which is supposed to provide active sites in OER and enhance the adsorption of OH$^-$ according to our DFT calculations[59]. Herein, the essential issue is to break or weaken the strong bond between μ-oxo and proton. The electric field enhanced OH$^-$ concentration contributes to the acceptance of decoupled protons and regulates the intermediates (*OH and *OOH).

This phenomenon indicates that the function of a gate voltage is beyond the field-effect-induced ion accumulation, the strong interaction between the adsorption and electronic states ultimately changes the energy profile of redox transition and charge transfer, like the self-gating phenomenon in semiconductors[60]. The nanowire in the electrolyte is inert when applied to a small overpotential, and the apparent resistance decreases slightly until overcome the energy barrier for OER, the state of MnO$_2$ nanowire shows an "open" state. The induced change of electrostatic potential ($\Delta\phi$) is determined by $\Delta\phi = \frac{en}{\varepsilon_r\varepsilon_0/d_{EDL}}$, where e, $n$, $\varepsilon_r$, $\varepsilon_0$, and $d_{EDL}$ represents the elementary charge, carrier concentration, the relative dielectric constant of the KOH electrolyte, vacuum dielectric constant and the thickness of the electrical double layer, respectively. When a much stronger electric field is applied, $\Delta\phi$ is estimated to be decades MV/cm, the induced high carrier concentration in nanowire results in the heavily doped surface region. Considering the adsorption states, the applied external electric field reorganises the dipole orientation in the μ-oxo-O(H), and weakens the ionic structure to cause the "soft homolysis"[28] for concerted PCET. The applied electric field provides an oriented chemical potential gradient and enhances the limited deprotonation process, contributing to a more efficient OER. Hence, the applied gate electrode realises an electrochemical FET and the mimicked electric-field-induced significant regulation of proton-electron coupling. This feature exerts the structural advantages of the biomimetic motif in α-MnO$_2$ and contributes to the coupled redox transitions and the enhanced charge transfer. Besides, the effect of the accompanying evolution of electronic states of semiconductor catalysts during the OER process is also an essential point of concern when investigating rational activity.

### External electric-field-assisted flow cell for water splitting

Although the electric-field-assisted oxygen evolution has been demonstrated, introducing an external electric field in practical water-splitting devices remains challenging. Normally, the alkaline electrolyser for water splitting is based on the stacked cell with the circulating electrolyte, in which the cathode, membrane and anode are stacked layer by layer. Based on this typical configuration, we propose to use the metal plate as the gate electrode to realise an electric-field-assisted overall water-splitting system. Herein, we set up a prototype external electric field enhanced anion exchange membrane (AEM) cell to

demonstrate the effectivity of the electric field based on a centimetre-sized electrode (Fig. 4a). The inner circuit supplies the voltage ($V_1$) for water splitting and the outer circuit supplies the gate voltage ($V_2$). In this way, an electric field is built vertically to the thin film electrode. According to a traditional plate capacitor model, due to the distance between two gate plates being approximately 1 mm, the enhancement strength is apparently lower than the single nanowire system. Hence, in this cell, the experimental gate voltage is accordingly increased approximately 10 times to 30 V. The polarisation curves of overall water splitting were measured at room temperature with a flow of 1 M KOH electrolyte (Fig. 4b). When applying a 30 V gate voltage, the current density at 2 V cell voltage is pronouncedly increased from 99 to 134 mA/cm$^2$ (34% increment). In this system, the commercial Pt/C was used as the cathode, which shows negligible overpotential for hydrogen evolution. Thus, the electric-field enhanced OER contributes to the pronouncedly optimised cell voltage (100 mV reduction of cell voltage at 100 mA/cm$^2$) for driving overall water splitting.

We further investigated the electric-field-enhanced water splitting by the chronoamperometry method, simulating the hydrogen production industry. Applying a constant cell voltage, the current density response was recorded during a staircase scanning gate voltage (Fig. 4c). The current density increased by ~25 mA/cm$^2$, compared to the state without gate voltage. And the gate leakage current density is maintained at around 1 mA/cm$^2$ (Supplementary Fig. 24). To give a quantified energy increase, we calculate the average power density of gate consumption and increase of electrolyser (Fig. 4d). Compared with the initial state, the power density of electrolyser at 30 V gate voltage increases by 62.2 mW/cm$^2$, while the gate consumption is kept around 17.5 mW/cm$^2$, which contributes to a 44.7 mW/cm$^2$ net increase. To verify the long-term working stability, the long-term galvanostatic test was performed by the flow cell with 30 V gate voltage (Fig. 4e). The initial cell voltage is 2.0 V and after 30 h, the cell voltage slightly increases to ~2.25 V, demonstrating a good stability in alkaline condition. To prove the universality of this field-enhanced strategy, we tested another proton-rich catalyst Ni(OH)$_2$ (Supplementary Fig. 25). The enhanced oxidation peaks of Ni$^{2+}$/Ni$^{3+}$ prove the enhanced deprotonation process, and the enhanced water splitting current at 30 V gate voltage demonstrates a similar enhancement mechanism. This result further shows a robust stimulating effect on overall water splitting by electric-field-enhanced OER. Building upon this system, we have demonstrated the reproducibility of the proposed field-assisted water-splitting process, which holds promise for reducing the cost of commercial hydrogen production. Additionally, given the capacity to adjust proton-electron reactions within a flow reaction system, we believe this strategy can be applied to various industrial catalysis systems.

As the structural analogy to the natural oxygen evolution complex, manganese oxide is a promising oxygen electrode catalyst, while OER activity is limited by the imbalanced protons incorporation in redox transitions. This inspires us to think that in addition to structural bionics, the regulation of acid-base chemistry reactivity and the reaction environment is also a potential strategy. In this work, we employed modelled α-MnO$_2$ to demonstrate the crucial role of di-μ-oxo bridged Mn for oxygen evolution reactions by in situ spectroscopic fingerprints. DFT calculations demonstrated the deprotonated surface is essential for maintaining dual sites adsorption of key intermediates *OOH to achieve a moderate adsorption energy. These results show that the bridging oxygen is the site for the lattice proton incorporated reaction, providing a moderate active site and energy profile for the following OER process. We then proved that this pathway could be enhanced by applying an external electric field, which improved the redox transition of MnO$_2$ and the deprotonation process. The increased deprotonation process contributed to the concerted proton-electron transfer, thereby improving the OER performance of MnO$_2$. By this means, a superior performance, 360 mV

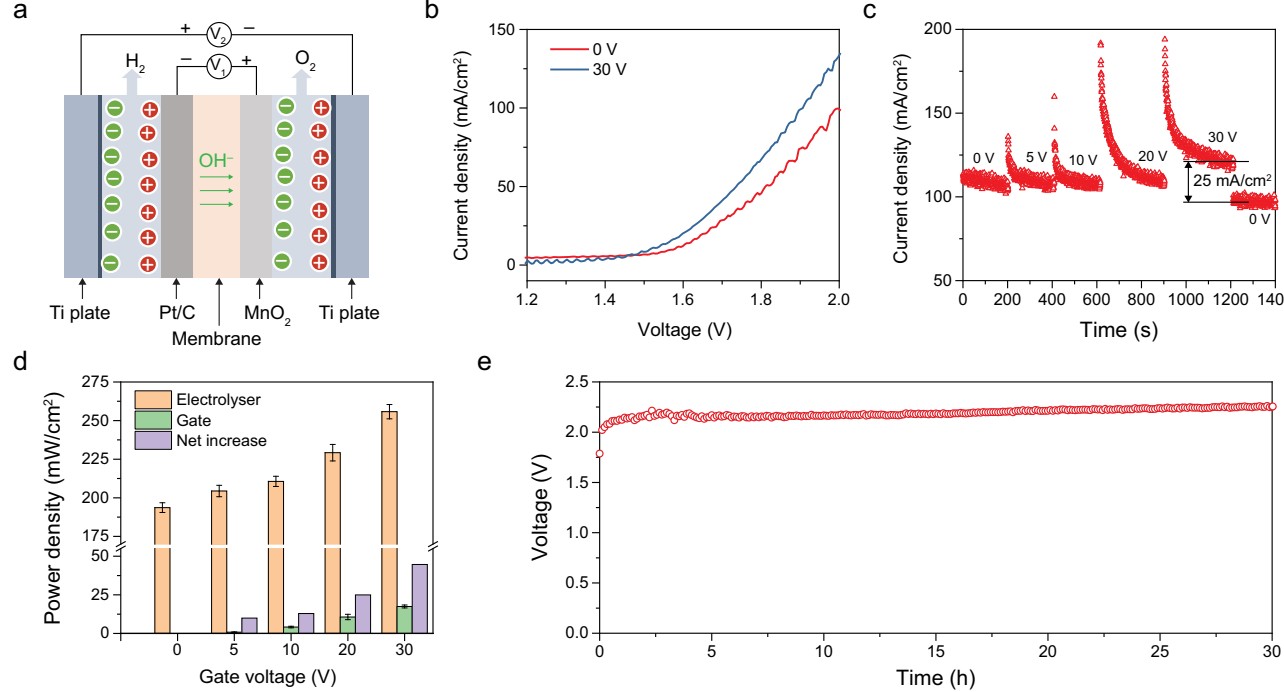

**Fig. 4 | The electrochemical performance of overall water splitting in an electric-field-assisted AEM cell. a** The schematic illustration of the external electric field enhanced anion exchange membrane cell with 1 M KOH electrolyte flow. The commercial Pt/C (20 wt%) was used as the cathode and the $MnO_2$ nanowire was the anode. The gate voltage $V_2$ was applied on the Ti plate (pre-oxidised), with a fluid channel and an oxide layer on the surface to eliminate the leakage current. The cell voltage $V_1$ was applied to the cathode and anode to drive the water splitting. **b** The polarisation curves of overall water splitting under different gate voltages.

**c** The plot of chronoamperometry response of electric-field-enhanced AEM cell under different gate voltages ($V_2$). The cell voltage was set at a constant voltage ($V_1$ = 2 V). **d** The bar charts of power density of electrolyser under different gate voltages, the corresponding power of gate consumption and the net increase of power density. The data are from Fig. 4c and Supplementary Fig. 24. The power density is divided by the membrane area. **e** The plot of long-term cell voltage at constant current density of 100 mA/cm² with 30 V gate voltage.

overpotential, and 56 mV/dec Tafel slope have been achieved. We conclude that the electric field changes the surface ligand field environment and controllably influences the redox transition of $MnO_2$. Hence, the adsorbate evolution mechanism and the charge transfer process are beneficially regulated. This effect was further proved by a prototype external electric-field-enhanced AEM cell, which shows a significant increase in current density and higher power density for overall water splitting. This flow cell model successfully demonstrates applying an external electric field to regulate electrochemistry and holds great promise in a wide range of applications such as electrosynthesis, electrolysis, electrodialysis, etc. We believe these results will contribute to understanding the OER mechanism and provide opportunities to produce new-generation electrochemical models.

## Methods

### Preparation of $MnO_2$ nanowire
In a typical preparation procedure, 2 mmol $KMnO_4$ and 2 mmol $NH_4F$ were added to 80 mL deionised water and magnetically stirred for 30 min at room temperature. Then the obtained solution was transferred into a 100 mL autoclave and kept at 180 °C for 24 h. The $MnO_2$ nanowires were obtained after washing and drying at 80 °C for 12 h in a vacuum atmosphere. The structure characterisations are presented in Supplementary Fig. 1 and Supplementary Notes.

### In situ Raman spectra measurement
The in situ Raman spectra were acquired by Horiba LabRAM HR Evolution with a He/Ne laser of λ = 633 nm and 4.9 mW. In a typical in situ Raman test, the grating line density is 600 gr/mm and acquisition time is 5 s and accumulates 4 times. The CV was performed by a CHI 760E electrochemical workstation at a scan rate of 0.5 mV/s in a customised

Teflon cell with 1 M KOH solution. A glassy carbon electrode (diameter = 3 mm) worked as the working electrode, an Ag/AgCl electrode (saturated potassium chloride solution, $E_{(RHE)} = E_{(Ag/AgCl)} + 1.01 V$) as the reference electrode, and a polished platinum wire as the counter electrode.

### Rutherford backscattering spectrometry and elastic recoil detection
Simultaneous ERD and RBS measurements were performed at the Surrey Ion Beam Centre using a 2 MeV $^4He^+$ beam generated by the 2 MV Tandetron accelerator. The ERD measurements were carried out using an incident angle of 73°, and recoiled H was collected using a 3 mm × 29 mm Si PIN photodiode mounted at a scattering angle of 30.4° with an 8 μm thick (±20%) Al range foil placed in front to filter out forward scattered ions (Supplementary Fig. 9a). RBS spectra were acquired simultaneously using two passivated implanted planar Si (PIPS) detectors placed at a scattering angle of 173.4° in Cornell geometry and 148.6° in IBM geometry[61]. Spectra were acquired for total integrated charges of 10 μC for each sample. Unfortunately, features in the spectra made it evident that the primary beam had not always exclusively hit each sample due to misalignment caused by the small size of the samples and spreading out of the beam at glancing angles. Corrections were therefore made to account for part of the beam hitting the sample holder plate.

### Fitting of RBS and ERD
The simulation and fitting of ERD and RBS measurements were carried out using the software SIMNRA[62]. SRIM2013 stopping powers were assumed for all elements. Built-in Rutherford scattering cross sections were used for all species except H (ERD) and O (RBS) which used Sigma

Calc values. Andersen screening was used throughout. Dead-time was not accounted for. Pile up used the SIMNRA fast model. Multiple scattering was not accounted for in the RBS spectra. Hence the fit in Supplementary Fig. 9 is lower than the experimental data at low energies (<0.3 MeV), where multiple scattering becomes important. We have not shown this region in the sample spectra so that the important higher energy part of the spectrum associated with the sample surface can be drawn at a suitable scale. Likewise, the increase in the ERD signal <0.1 MeV for the Al sample plate can be attributed to forward scattered He that was not stopped by the thinner portions of the 8 μm range foil. The two RBS spectra and ERD spectra for each sample were fitted using a bi-layer model. The layer compositions were adjusted so the fits to all three spectra were satisfactory, judged by eye. SIMNRA assumed the samples consisted of two smooth layers. The nano-rod structure accounts for the poor fits on the Mn edge (~1.5 MeV). Samples of $Si_3N_4$ implanted with 3 atomic % H and Kapton with a 30 nm thick Al backing were used for the solid angle and energy calibrations for the RBS and ERD detectors.

The incident ion beam clipped the sample plate during the 0.9 V and 1.6 V measurements as evident from the Al and Cu edges and reduced Mn and O intensities. Simultaneous ERD and RBS spectra were collected for the naked sample plate (Supplementary Fig. 9b, c). The fractions of the total 10 μC charge that hit the sample and sample plate could then be determined by fitting the RBS spectra Fig. 2e and f with a combination of the plate and a manganese dioxide spectrum. This was then used to normalise the collected H ERD spectra (Fig. 2e, f) as if they had been impinged by the full 10 μC. A (relatively small) correction was also made to each ERD spectrum (Fig. 2e, f) to account for the H present in the sample plate. It was determined that 60 % of the beam hit the sample plate for the 0.9 V sample and 20 % for the 1.6 V sample.

### Fabrication of single MnO₂ nanowire device

The obtained $MnO_2$ nanowires were diluted and dispersed in the ethanol and followed by a low-speed spin-coating to transfer nanowires to the silicon substrate (with a 300 nm oxide layer). Then the microelectrodes (Ti/Au, 10/100 nm) were fabricated by electron-beam lithography (EBL) (JC Nabity Lithography Systems, Nanometre Pattern Generation System) and physical vapour deposition (Kurt J. Lesker, PVD75). At last, SU-8 2002 photoresist (MicroChem Corp.)/$Si_3N_4$ layer was used as an insulating layer fabricated by EBL and magnetron sputtering deposition system (PDVACUUM, PD-200C).

### Electrochemical measurement

In a typical individual nanowire-based three-electrode system, 1 M KOH (pH = 13.8) was used as an electrolyte and a graphite rod and a micro Hg/HgO electrode (1 M KOH) were used as counter electrode and reference electrode, respectively. The potentials were calibrated to RHE at 298 K ($E_{(RHE)} = E_{(Hg/HgO)} + 0.91$ V). The OER performance of individual nanowire at different gate voltages was tested by combining the electrochemical workstation with the probe station and semiconductor device analyser. The polarisation curves and CV curves were measured at a scan rate of 5 mV/s, without IR drop calibration.

### Instrument and material characterisation

Crystallographic information was collected by Bruker D8 Discover X-ray diffractometer with a non-monochromate Cu Kα X-ray source (tube current: 40 mA, tube voltage: 40 kV). Raman spectra were obtained by Horiba LabRAM HR Evolution. The FESEM image was recorded using a JEOL JSM-7001F microscope at an acceleration voltage of 20 kV. The STEM images were collected in a CEOS probe corrected FEI Themis TEM with 300 kV accelerating voltage and the cross-sectional MnO₂ nanowire sample was prepared by a dual-beam FIB (FEI Helios Nanolab G3). AFM images were measured by AIST-NT SmartSPM 1000 Scanning Probe Microscope. XPS analysis was performed by Thermo Fisher Scientific ESCALAB 250Xi XPS System with Al Kα source. The binding energy was corrected by the C 1s peak (284.6 eV) for the adventitious carbon.

### External electric field enhanced anion exchange membrane cell

In this AEM cell, the electrodes are directly connected to the gas diffusion layer with the catalysts layer working as inner electrodes for water splitting. The outer electrodes were connected to titanium (Ti) plates working as the outer gate electrode. The Ti plate was pre-oxidised in 1 M KOH at 30 V for 30 min to form an oxide layer to minimise the leakage current. The electrolyte is 1 M KOH flow driven by a peristaltic pump. Nickel foam was used as a gas diffusion layer as well as conducting electrode to avoid oxidation of traditional carbon electrodes. The catalyst layer on gas diffusion was prepared by the spray method with 1 mL catalyst ink on 1 cm² Ni foam. The catalyst ink in this work is prepared in the same proportion, with 5 mg catalyst, 5 mg carbon powder, 750 μL isopropanol, 200 μL water and 50 μL Nafion. The commercial Pt/C (20 wt%, Aldrich) was used as cathode and the $MnO_2$ nanowires were the anode catalyst.

### Reporting summary

Further information on research design is available in the Nature Portfolio Reporting Summary linked to this article.

### Data availability

Source data are provided with this paper.

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

## Acknowledgements

L.M. acknowledges support from the National Key Research and Development Program of China (2020YFA0715000). Y.Z. acknowledges support from Engineering and Physical Sciences Research Council (EPSRC, EP/V002260/1) and UK National Ion Beam Centre (Project: 562 Grant 1). X.P. acknowledges the Fundamental Research Funds for the Central Universities (WUT: 2020-YB-014) and support from the China Scholarship Council (CSC) and University of Oxford. N.P. acknowledges support from Royal Society International Exchanges Scheme (IEC\NSFC \211298). The authors thank the beamline BL11B of Shanghai Synchrotron Radiation Facility (SSRF) for in situ XAS experiments. The authors thank Prof. Bruce Dunn from University of California Los Angeles and Dr. Tim Rosser from UK National Physical Laboratory for helpful discussions and useful suggestions.

## Author contributions

X.P., M.Y., Y.Z. and L.M. conceived the concept and designed the experiments; X.P. carried out the main experiments with X.Z. and J.Z.; Q.L. and X.L. performed the DFT calculation. C.S. carried out STEM characterisation. C.M., P.C., M.K.S., R.S., N.P. and J.E. conducted the ion-beam characterisation and analysis; X.P., M.Y., S.C.E.T., Y.Z. and L.M. analysed the data and wrote the manuscript. All authors discussed the data and commented on the manuscript.

## Competing interests

The authors declare no competing interests.
