## [Peer Review File · Nature Communications]

Electric-field-assisted Proton Coupling Enhanced Oxygen Evolution ReactionREVIEWER COMMENTS

Reviewer #1 (Remarks to the Author):

This manuscript a facile electric-field-assisted is applied to promote the proton-electron transfer for advancing OER. It identifies the site of the lattice proton merging reaction and explores the reasons for the improvement of OER properties by applying an external electric field. The findings contribute significantly to the progress of an in-depth understanding of the role of proton-incorporated redox transition. However, there are still some problems to be solved. After minor revisions, this article can be accepted for publication, please refer to the following for specific comments:

Q1: The curve in Figure S11(c) is considered as "This means the approximate combination of zero-order dependence and inverse first-order dependence on H^+ during a multi-step oxygen reaction sequence" only by giving three points, which is too absolute. Especially, this conclusion is used on page 16 of the manuscript. More tests should be carried out on the relationship between potential and pH and an approximate fitting equation should be given.

Q2: The Rutherford backscattering spectroscopy spectra at 0.9 V and 1.6 V are given on page 11, but the conclusion is that protons behave differently at 1.1 V and 1.2 V, and the description here is vague.

Q3: "The shift of open-circuit potential can be attributed to the concentrated anions (OH^-) at the surface of the nanowire, and the injected electrons adjusted the Fermi level" on Page 16, it is necessary to provide corresponding characterization to prove the change of Fermi energy level.

Q4 : It is necessary to supplement the performance test at 10 mA cm^{-2} and 50 mA cm^{-2} . Additionally, the stability of catalysts at high current densities is crucial, requiring long-term cyclic stability testing.

Reviewer #2 (Remarks to the Author):

Pan et al. report a study of OER over a-MnO₂ nanowires. They show that under an external (gate) electric field, which is separated from the working electrode by an insulator, the electrochemical reaction is enhanced by reducing the overpotential from 440 mV to up to 360 mV. They attribute this effect to a change in the concentration of OH^- , which promotes a concerted proton-electron transfer instead of a sequential one. However, such statement is not fully supported in the text and it seems to me as a hypothesis rather than a conclusion. Another issue is that it is not clear how the gate electric field is able to affect so much the working electrode since it is separated by an insulating layer (in contrast to ref. 51, where the molecules directly "feel" an oriented electric field). Since I am a computational chemist, and I will assess the computational part of the manuscript, leaving the core idea to expert Reviewers in the electrochemistry field.

My major concern is the great disconnect between the experiments and the DFT calculations. I cannot see how DFT contributes to the experimental work. There is no discussion in the main text and the section in the supplementary material is difficult to follow.

-Experiments suggest that the mechanism changes from sequential to concerted proton-electron transfer. This major feature of the main text is not present in the calculations.

-Experiments suggest that the gate electric field generates a high concentration of proton acceptors, OH^- . If I understood correctly, this would suggest that the deprotonated structure (MnO₂) is predominant since there are more proton acceptors. But the computed overpotential of (MnO₂) is larger than that of protonated (MnO₂+4H).

-The lattice oxygen mechanism has not been considered.

-Raw structures must be reported for reproducibility and visualization. It is very difficult to evaluate any computational work just by looking at numbers in profiles (Figure S9). The convoluted description of bonding (page S6) would be easier to understand if structures/schemes are provided.

There are other technical issues I would like to rise.

- I assume they performed spin-polarized calculations, but this is not indicated in the text. Also, magnetic moments for relevant Mn and O atoms should be reported.
- It would be useful to cite previous DFT+U work to support the choice of $U = 4.5$ V for this material.
- Please report details about the numerical computation of vibrational frequencies.
- Figure S9a: When they say "proton", I assume they mean "hydrogen" since charges cannot be described in PBC. Then, when adding/removing H, what are the oxidation states of Mn atoms in each configuration? How can they affect the overpotential or correlate with the potential determining steps?
- Figure S9b: It is very difficult to discern the tonalities of blue. Please change the color palette or include the numerical values in the graph.
- Nomenclature: It is "potential determining step" rather than "rate determining step". Also, sometimes they refer to "barriers" but transition states are not computed, so this is not entirely correct.
- Figure S10a: typos in X axis, it should be "site" instead of "sitie".
- Figure S11c: They present a regression line with only three points.

Reviewer #3 (Remarks to the Author):

This study reports a concerted proton-electron transfer strategy for enhancing the OER activity of α -MnO₂ through the apply of an external electric field during the reaction process. This work is well done with a coherent logic flow, the in situ study is conducted in detail, and the key finding is interesting. However, it lacks sufficient novelty as the protonation/deprotonation processes of the prototype α -MnO₂ material have been well-established in Zn-MnO₂ batteries, and the conclusions need to be strengthened by more rational experimental and characterization designs. In addition, although the water electrolysis employing the concept developed in this study has been successfully demonstrated, its practical competitiveness remains an issue. Before its publication, the following concerns should be well addressed.

1. I am wondering whether the in situ Raman spectra can be accurately recorded at specific voltages using the CV technique, why not using the chronoamperometry technique at a certain constant potential?
2. If there is bubble interference at 1.7 V, then the Raman spectrum recorded at 1.7 V can be excluded from Fig. 1d, as it is not relevant to the subsequent discussion.
3. What's the reason for the insignificant changes of the peaks at the first cycle from 1.0-1.7 V compared with those in the second cycle within the same potential window as shown in Fig. 1d?
4. In the illustration of the in situ Raman spectra, the authors claim that "after the anodic process, the profiles return to the typical doublet vibration bands". However, this conclusion is not very persuasive, as at the initial stage of the cathodic process, this peak is even invisible at 1.6 V.
5. Could the author explain the different peak shifts of the ν_2 peak within the potential range of 1.2-1.0-1.2 V as shown in Fig. 1f? In my opinion, it is not a reversible fluctuation as claimed in the manuscript.
6. To better support the structural evolution, I suggest to conduct the in situ XAS study in a similar potential trip as that for in situ Raman spectra.
7. In Fig. 1g, the Mn-O bond length exhibits a slight increase from 0.9 to 1.3 V, rather than showing a decrease as stated. What's more, what's the reason for the enhanced scattering intensity of the Mn-O bond with the increasing potential?
8. There is an issue regarding the transformation from a double octahedra configuration to the partial tetrahedra configuration as depicted in Fig. 1h. When one of the protonated bridge O (di- μ -oxo-O) that shared by two octahedra detached from another, the octahedra remain intact (i.e. the coordination number of O, etc.).
9. The possibility of this partial transformation from octahedra to tetrahedra contributing to a typical lattice-O involved OER mechanism should be considered. As an obvious pH-dependent OER activity is observed in this study, the authors are encouraged to elucidate the pH effect in depth.

10. The role of the accommodated K^+ needs to be clarified, and it is also necessary to investigate whether K^+ cations in the electrolyte compete with protons for adsorption on MnO_2 surface.
11. What's the proton source for the protonation of MnO_2 in an alkaline medium of 1 M KOH? And as shown in Fig. 1, the protonation occurs at the cathodic process, while the OER runs at anodic process. Moreover, how to ensure its continuous supply during the OER process, i.e., maintaining a dynamic incorporation and deprotonation process in enhancing the OER?
12. The economic issue of the overall water splitting using an external electric field with a high voltage remains. As it lacks evident advantage compared to other water electrolysis systems, particularly those capable of delivering ampere-level current density at low cell voltages.
13. Several supplementary figures are not mentioned in the manuscript, such as supplementary Fig. 1 a-c.
14. There exists a few typo errors or inappropriate expressions, the whole manuscript should be further polished.

Response letter to the reviewers' comments

Manuscript ID: #NCOMMS-23-59958-T

Response to Reviewer#1

This manuscript a facile electric-field-assisted is applied to promote the proton-electron transfer for advancing OER. It identifies the site of the lattice proton merging reaction and explores the reasons for the improvement of OER properties by applying an external electric field. The findings contribute significantly to the progress of an in-depth understanding of the role of proton-incorporated redox transition. However, there are still some problems to be solved. After minor revisions, this article can be accepted for publication, please refer to the following for specific comments.

Response: We thank the Reviewer for the passionate recognition of the innovation and significance of our work. The detailed revisions are presented in the following response.

1. The curve in Figure S11(c) is considered as “This means the approximate combination of zero-order dependence and inverse first-order dependence on H⁺ during a multi-step oxygen reaction sequence” only by giving three points, which is too absolute. Especially, this conclusion is used on page 16 of the manuscript. More tests should be carried out on the relationship between potential and pH and an approximate fitting equation should be given.

Response: We greatly appreciate the insightful feedback from the reviewer. Upon revisiting our experimental data, we recognised that the initial results may have suffered from a suboptimal linear relationship, possibly due to the decreased ionic strength of electrolyte. To address this, we conducted further electrochemical measurements in KOH electrolyte across various pH levels, using potassium sulphate as a supporting electrolyte to maintain a consistent concentration of K⁺ ions.

The updated pH-dependent electrochemical analysis is presented in Figure R1. We recalculated the reaction order (ρ) using the following expression:

$$\rho = \left(\frac{\partial \log j}{\partial \text{pH}} \right)_E = - \left(\frac{\partial E}{\partial \text{pH}} \right)_i / \left(\frac{\partial E}{\partial \log j} \right)_{\text{pH}}$$

The new data reveals a clear first-order reaction with a fitted slope of 1.03 ± 0.06 . This suggests a reverse first-order dependence on H⁺, indicating that the oxygen evolution reaction (OER) is significantly influenced by the concentration of the H⁺ acceptor, namely OH⁻.

In light of these new findings, we conclude that the OER process on the MnO₂ surface exhibits an inverse first-order dependence on H⁺ concentration. The manuscript has been updated accordingly to reflect these findings on pages 15 and 16, and the revised data in Supplementary Fig. 17, with changes highlighted in red for clarity.

Figure R1. (a) pH dependence for CV curves of MnO₂ in KOH solution with different concentrations. (b) The relationship between current density at 1.8 V vs. RHE and pH.

Revised main text: (pages 15 and 16) “To understand the PCET process of OER, the OER activity of MnO₂ at different pH was measured (Supplementary Fig. 17a,b). Reaction order (ρ) can be determined by the linear relationship between $\log j$ and pH, $\rho = \left(\frac{\partial \log j}{\partial \text{pH}} \right)_E = - \left(\frac{\partial E}{\partial \text{pH}} \right)_i / \left(\frac{\partial E}{\partial \log j} \right)_{\text{pH}}$, where j is current density, E is potential versus RHE. The fitted slope value is 1.03 ± 0.06 , presenting the reverse first-order dependence on H^+ . Hence, the rate-determine step of α -MnO₂ for OER is decided by the concentration of H^+ involved in reaction, resulting in a strong pH dependence OER activity (i.e., decoupled proton-electron transfer). CV curves at different pH also provide some information about the redox transition of Mn (Supplementary Fig. 17c,d). The separation of oxidation and reduction peak potential shows a dependence on pH and scan rate. With the increase of pH, the width of CV peaks and the redox potential separation decreases at the same scan rate, demonstrating a decreased polarisation of proton-electron reaction at resting state before OER. The results above demonstrate the OER process on the MnO₂ surface is an uncoupled proton-electron transfer reaction, which shows inverse first-order dependence on H^+ concentration. This also indicates that increasing the H^+ acceptor can adjust the electron and proton coupling to enhance OER. Herein, the energy profile of the OER process is dominated by the proton configuration, and maintaining a circulation of lattice and surface protons will contribute to moderate adsorption energy OER thermodynamics.”

Revised Supplementary Fig. 17:

Supplementary Figure 17. pH-dependent OER measurement. (a) pH dependence for CV curves of MnO₂ in KOH solution with different concentrations. The electrolyte was prepared by adding potassium sulphate to maintain the constant K⁺ strength. (b) The relationship between current density at 1.8 V vs. RHE and pH. (c) The CV curves of MnO₂ were measured at different pH KOH solutions with different scanning rates. (d) The corresponding peak separation at different scan rates. $E_{\text{peak}} - E_{\text{eq}}$ represents the difference between peak potential and equilibrium potential.

2. The Rutherford backscattering spectroscopy spectra at 0.9 V and 1.6 V are given on page 11, but the conclusion is that protons behave differently at 1.1 V and 1.2 V, and the description here is vague.

Response: We appreciate the reviewer's attention to the detail and precision of our descriptions regarding proton behaviour at specific potentials. We acknowledge that the initial explanation may not have sufficiently connected the Rutherford backscattering spectroscopy (RBS) data at 0.9 V and 1.6 V vs. RHE to our conclusions about proton behaviour at 1.1 V and 1.2 V.

To clarify, electrochemical measurements identified ~1.05 V vs. RHE as a critical redox peak, where the potential above favours Mn site oxidation (deprotonation) and below favours reduction (protonation). We selected 0.9 V and 1.6 V for RBS measurements to represent fully protonated and deprotonated states of MnO₂, respectively, thereby framing the entire range of the protonation-deprotonation process. Additionally, *in situ* Raman spectroscopy provided insight into the gradual transition of these states, with noticeable changes occurring around the redox centre at 1.05 V vs. RHE. These combined results guided our choice of potentials for detailed hydrogen concentration analysis.

To better communicate these points, we have revised the relevant descriptions on page 11 to explicitly link the chosen potentials for RBS measurements to the observed electrochemical behaviour and *in situ* Raman spectroscopy findings. These changes have been highlighted in red to aid in their identification.

Revised main text: (page 12) “We compared two set samples conditioned at 0.9 and 1.6 V vs. RHE to ensure that complete protonated and deprotonated states were measured. The RBS spectra of the pristine samples show that α -MnO₂ composition fits the spectra well.”

3. “The shift of open-circuit potential can be attributed to the concentrated anions (OH^-) at the surface of the nanowire, and the injected electrons adjusted the Fermi level” on Page 16, it is necessary to provide corresponding characterization to prove the change of Fermi energy level.

Response: We thank the reviewer for the suggestion to provide characterisation on the change in the Fermi level. In our experimental setup, *in situ* measurements of the Fermi level are significantly challenging due to the dynamic nature of the electrochemical environment, where constant fluctuations in ion concentrations and potential gradients can obscure the precision of Fermi level assessments. Additionally, the integration of appropriate spectroscopic techniques, such as photoemission spectroscopy, within the operational electrochemical cell without interfering with the MnO_2 nanowire activity or the integrity of the measurements presents substantial experimental difficulties. These complexities are compounded by the need to maintain a stable reference state amidst the constantly evolving electrochemical reactions at the nanowire surface. Although trying our best to design a system compatible with *in situ* measuring Fermi level, it remains technically difficult to make it.

Figure R2. (a) Energy diagrams of MnO_2 electrochemical system without applied back gate voltage. (b) Energy diagrams of MnO_2 electrochemical system applied positive back gate voltage. The symbols in diagrams are vacuum level (E_{vac}), Fermi level of back gate (E_G), work functions of back gate (Φ_G) and Reference electrode (Φ_{ref}), electron affinity of MnO_2 (χ), Fermi level offset ($\delta = E_c - E_f$). Felectrical double layer (EDL), electrode potential (V_E), and vacuum level shifts in SiO_2 ($\Delta\phi_G$). (c) The schematic illustration of back gate electric field. The "+" and "-" represent positive and negative charge carrier, respectively. (c) The plot of open-circuit potential versus back gate voltage.

On the other hand, shifts in open-circuit potential are indicative of changes in the Fermi level shift within an electrochemical system. To elucidate this further, we present an analysis of open-circuit potential variation. Figures R2a and R2b depict the energy level diagrams of the MnO_2 electrochemical system without and with an applied back gate voltage, respectively. When applied a positive gate voltage, the energy level shifts due to the charge accumulation at the interface, following Poisson's equation. In our case, a large gate electrode works on both channel material (MnO_2 nanowire) and electrolyte. The back gate energy level shift ($\Delta\phi_G$) is charging through polarisation of the insulating layer (SiO_2) and the electrical double layer shift ($\Delta\phi_{\text{EDL}}$) is through anion (OH^-) accumulation on the electrode surface charging the double layer. Based on the energy level diagram, Fermi level shift can be estimated by the difference of relative offset to conduction band bottom, $\Delta\delta = \delta_0 - \delta = e(V_E - V_{E0})$. The overall shift can be calculated by the charge coupling with back gate and electrical double layer. The total charge Q_w is expressed as $Q_w = C_G V_G + C_{\text{EDL}} V_E$, which indicates a linear relationship between V_E and V_G , $\frac{\partial V_E}{\partial V_G} = -\frac{C_G}{C_{\text{EDL}}}$, where C_G and C_{EDL} represent gate and EDL capacitance. Figure

R2c shows the linear relationship between gate voltage and electrode potential and the slope is fitted to be -0.6 . Hence, $\Delta\delta = 0.6e * V_G$. Based on this equation, we can estimate the shift of Fermi level relative to the conduction band. Herein, the open-circuit potential shift is a direct evidence of Fermi level shift.

In summary, the shift observed in the open-circuit potential serves as a proxy for the Fermi level shift. The detailed figures and analyses have been integrated into the main text, and the revisions have been highlighted on pages 18 and 19, in Fig. 3f, and Supplementary Fig. 22.

Revised main text: (pages 18 and 19) “Hence, the equilibrium is considered to be *saturated with the concentrated proton acceptor*. Fig. 3f shows diagram illustration of energy level with positive gate voltage. When applied a positive gate voltage, the energy level shifts due to the charge accumulation at the interface, following Poisson's equation. In our case, a large gate electrode works on both channel material (MnO_2 nanowire) and electrolyte. The back gate energy level shift ($\Delta\phi_G$) is charging through polarisation of the insulating layer (SiO_2) and the electrical double layer shift ($\Delta\phi_{EDL}$) is through anion (OH^-) accumulation on the electrode surface charging the double layer. Based on the energy level diagram, Fermi level shift can be estimated by the difference of relative offset to conduction band bottom, $\Delta\delta = \delta_0 - \delta = e(V_E - V_{E0})$. The overall shift can be calculated by the charge coupling with back gate and electrical double layer⁵⁸. The total charge Q_w is expressed as $Q_w = C_G V_G + C_{EDL} V_E$, which indicates a linear relationship between V_E and V_G , $\frac{\partial V_E}{\partial V_G} = -\frac{C_G}{C_{EDL}}$, where C_G and C_{EDL} represent gate and EDL capacitance. Supplementary Fig. 22 shows the linear relationship between gate voltage and electrode potential and the slope is fitted to be -0.6 . Consequently, we derived a numeric expression of energy shift induced by gate voltage $\Delta\delta = 0.6eV_G$.”

Revised Figure 3:

Figure 3. The electrochemical performance of the single $\alpha\text{-MnO}_2$ nanowire device. **a**, The schematic diagram of the single nanowire electrocatalytic device where a single $\alpha\text{-MnO}_2$ nanowire is connected to the Au microelectrodes with Si_3N_4 as the insulating layer. **b**, Polarisation curves and **c**, Tafel plots of the single $\alpha\text{-MnO}_2$ nanowire at different gate voltages. Inset: the schematic illustration of the working principle of gate voltage. V_1 represents the potential applied to the working electrode and V_2 is the gate voltage. **d**, The statistics results of overpotential and Tafel slope plots at different gate voltages. The error bars represent the standard errors. **e**, The gate voltage-tuned open circuit potential of the single $\alpha\text{-MnO}_2$ nanowire. **f**, Energy diagrams of MnO_2 electrochemical system applied positive

back gate voltage. The symbols in the diagrams are vacuum level (E_{vac}), Fermi level of the back gate (E_G), work functions of the back gate (Φ_G) and Reference electrode (Φ_{ref}), electron affinity of MnO_2 (χ), electrical double layer (EDL), and vacuum level shifts in SiO_2 ($\Delta\phi_G$).

Revised Supplementary Fig. 22:

Supplementary Figure 22. The analysis of energy level alignment. (a) Energy diagrams of MnO_2 electrochemical system without applied back gate voltage. The symbols in diagrams are: vacuum level (E_{vac}), Fermi level of back gate (E_G), work functions of back gate (Φ_G) and Reference electrode (Φ_{ref}), electron affinity of MnO_2 (χ), Fermi level offset ($\delta = E_c - E_f$). Felectrical double layer (EDL), electrode potential (V_E), and vacuum level shifts in SiO_2 ($\Delta\phi_G$). (c) The schematic illustration of back gate electric field. The "+" and "-" represent positive and negative charge carrier, respectively. (b) The plot of open-circuit potential versus back gate voltage.

New reference:

58. Wang, Y. & Frisbie, C. D. Four-terminal electrochemistry: a back-gate controls the electrochemical potential of a 2D working electrode. *J. Phys. Chem. C* 128, 1819–1826 (2024).

4. It is necessary to supplement the performance test at 10 mA cm^{-2} and 50 mA cm^{-2} . Additionally, the stability of catalysts at high current densities is crucial, requiring long-term cyclic stability testing.

Response: We thank the reviewer for the suggestion to evaluate the catalyst's performance and stability at specific current densities. Recognising the importance of stability under practical operating conditions, we have now included a long-term stability test at a current density of 100 mA/cm^2 . This test, conducted in a flow cell with a 30 V gate voltage, offers a rigorous assessment of stability beyond the requested 10 mA/cm^2 and 50 mA/cm^2 tests. The initial cell voltage of 2.0 V only increased to approximately 2.25 V after 30 hours, affirming the catalyst's robust stability in alkaline conditions.

While we understand the reviewer's request for data at 10 mA/cm^2 and 50 mA/cm^2 , we believe that the data at 100 mA/cm^2 provides a more substantial demonstration of durability, as it subjects the catalyst to a higher stress test. Nevertheless, we are prepared to conduct additional tests at 10 mA/cm^2 and 50 mA/cm^2 if deemed necessary.

The results of the long-term stability test at 100 mA/cm^2 are documented in the Supplementary Information. The corresponding revisions have been made to pages 22 and Fig. 4e, with the changes marked in red.

Figure R3. The plot of long-term cell voltage at constant current density of 100 mA/cm^2 with 30 V gate voltage, highlighting the minimal increase in cell voltage over an extended period, indicative of the catalyst's high stability.

Revised main text: (Page 22) "To verify the long-term working stability, the long-term galvanostatic test was performed by the flow cell with 30 V gate voltage (Fig. 4e). The initial cell voltage is 2.0 V and after 30 h , the cell voltage slightly increases to $\sim 2.25 \text{ V}$, demonstrating a good stability in alkaline condition."

Revised Fig. 4:

Figure 4. The electrochemical performance of overall splitting in an electric field-assisted AEM cell. **a**, The schematic illustration of the external electric field enhanced anion exchange membrane (AEM) cell with 1 M KOH electrolyte flow. The commercial Pt/C ($20\text{wt}\%$) was used as the cathode and the MnO_2 nanowire was the anode. The gate voltage V_2 was applied on the Ti plate (pre-oxidised), with a fluid channel and an oxide layer on the surface to eliminate the leakage current. The cell voltage V_1 was applied to the cathode and anode to drive the water splitting. **b**, The polarisation curves of overall water splitting under different gate voltage. **c**, The plot of chronoamperometry response of electric field-enhanced AEM cell under different gate voltage (V_2). The cell voltage was set at a constant voltage ($V_1 = 2 \text{ V}$). **d**, The bar charts of power density of electrolyser under different gate voltages, the corresponding power of gate consumption and the net increase of power density. The data are from Fig. 4c and Supplementary Fig. 24 The power values are calculated by subtracting the initial output power density ($V_2 = 0 \text{ V}$) and power density is divided by the membrane area. **e**, The plot of long-term cell voltage at constant current density of 100 mA/cm^2 with 30 V gate voltage.

Response to Reviewer#2

Pan et al. report a study of OER over α -MnO₂ nanowires. They show that under an external (gate) electric field, which is separated from the working electrode by an insulator, the electrochemical reaction is enhanced by reducing the overpotential from 440 mV to up to 360 mV. They attribute this effect to a change in the concentration of OH⁻, which promotes a concerted proton-electron transfer instead of a sequential one. However, such statement is not fully supported in the text and it seems to me as a hypothesis rather than a conclusion. Another issue is that it is not clear how the gate electric field is able to affect so much the working electrode since it is separated by an insulating layer (in contrast to ref. 51, where the molecules directly “feel” an oriented electric field). Since I am a computational chemist, and I will assess the computational part of the manuscript, leaving the core idea to expert Reviewers in the electrochemistry field. My major concern is the great disconnect between the experiments and the DFT calculations. I cannot see how DFT contributes to the experimental work. There is no discussion in the main text and the section in the supplementary material is difficult to follow.

Response: We appreciate the reviewer's feedback concerning the underlying principles of our gate-voltage application and the connection between our experimental work and DFT calculations. The responses to these comments are below.

1. In this work, we first employed *in situ* spectroscopy to determine the role of protons in the OER process of MnO₂ and hypothesise that an external electric field can modulate this proton coupling. Figure R4a,b illustrates the device's energy diagrams without and with an applied positive gate voltage. Our experiments show a linear relationship between the gate voltage and OER activity, substantiated by both nano-scaled on-chip devices and millimetre-scaled flow cells, demonstrating the effectiveness of the external field. We acknowledge the complexities in the detailed mechanisms described in our initial manuscript, thus we revised the manuscript to provide a clearer and more solid rationale for the observed phenomena.

2. In our work, we fabricated a field effect transistor-like device to realise the field effect. In the reference, the electric field is applied to polarise a molecule to promote a specific reaction pathway. In our case, a gate electrode is set to apply an electric field, where the dielectric layer forms an oriented electric field. This configuration is the same as a semiconductor field effect transistor. When a positive gate voltage, the charge separation in silicon oxides induces the charge coupling in channel material (MnO₂) and the anion concentrated in an electrical double layer (EDL). The back gate energy level shift ($\Delta\phi_G$) is charging through polarisation of the insulating layer (SiO₂) and the electrical double layer shift ($\Delta\phi_{EDL}$) is through anion (OH⁻) accumulation on the electrode surface charging the double layer. This is the basic working principle of our field-tuned electrochemical device.

3. In terms of DFT calculations, we aim to reveal the difference between OER process in photo system II (PSII) and MnO₂ system Figure R4d,e. In both systems, the Mn sites go through valence cycling accompanied by OH⁻ adsorption and deprotonation. One of the important findings in DFT calculation is the adsorption states on MnO₂ surface. As is shown in the schematic illustration of PSII core complex (Mn₄CaO_x), O-O coupling was found to form an oxo motif between two Mn sites. This phenomenon was also observed in α -MnO₂, *OOH is absorbed on adjacent two Mn sites. This confirms the structural similarity between α -MnO₂ and Mn₄CaO_x. We further found that the proton on adjacent Mn sites can inhibit the formation of oxo motif which causes a large energy barrier. In summary, our DFT calculation results provide insight into the role of proton cycling in the OER process, which is essential for investigating catalysis activity-enhancing strategies.

In summary, after investigating more details from DFT calculations, we found important evidence to support the experiments and give a more in-depth analysis. Thank the reviewer again for the valuable suggestions. We believe these revisions strengthen our conclusions and make the work systematic.

Figure R4. (a) Energy diagrams of MnO_2 electrochemical system without applied back gate voltage. (b) Energy diagrams of MnO_2 electrochemical system applied positive back gate voltage. The symbols in the diagrams are vacuum level (E_{vac}), Fermi level of the back gate (E_G), work functions of the back gate (Φ_G) and Reference electrode (Φ_{ref}), electron affinity of MnO_2 (χ), electrical double layer (EDL), and vacuum level shifts in SiO_2 ($\Delta\phi_G$). (c) The schematic illustration of the back gate electric field. The "+" and "-" represent positive and negative charge carriers, respectively. (d) The S states in the oxygen-evolution reaction. The oxygen-evolving complex is photo-oxidized through a series of S states to produce molecular oxygen from water. In the final steps before $\text{O}=\text{O}$ bond formation, a new oxygen, O6, binds to the vacant site at Mn1. After a final photo-oxidation event, O5 and O6 appear poised to form an $\text{O}=\text{O}$ bond, releasing molecular oxygen, reducing the cluster, and beginning the catalytic cycle anew. Glutamic acid at position 189 is noted as E189. This figure is adapted from *Science*, 2019, 366,305-306. (e) The reaction cycle of MnO_2 for OER reaction.

1. Experiments suggest that the mechanism changes from sequential to concerted proton-electron transfer. This major feature of the main text is not present in the calculations.

Response: We are grateful for the reviewer's critical analysis and agree that our manuscript would benefit from a more explicit correlation between our DFT calculations and the experimentally suggested shift from sequential to concerted proton-electron transfer mechanisms. Addressing the complexity of simulating a complete concerted proton-electron transfer mechanism via DFT is indeed a formidable task, primarily due to the following factors:

Reaction Complexity: Concerted proton-electron transfers involve simultaneous transfer of electrons and protons across a complex energy landscape, which is difficult to capture fully in static DFT calculations that typically model discrete states.

Computational Limitations: The computational cost of accurately modelling the entire reaction pathway, including all possible intermediate states and transition states associated with PCET, can be prohibitive, especially when considering the dynamic nature of the electrochemical environment.

Despite these challenges, we have strived to enhance our computational models to capture the essence of this mechanism. Our DFT studies have focused on characterising the proton adsorption motifs and related energy profiles on the MnO₂ surface, which are pivotal in determining the OER pathway. The free energy diagrams presented in Figure R5 illustrate how the surface and tunnel proton configurations impact the various Gibbs free energy changes (ΔG_1 through ΔG_4) associated with the OER steps.

Specifically, our revised DFT models (Figure R6) address different proton configurations and their influence on the adsorption of *OOH intermediates. The models reveal that configurations without surface protons allow for *OOH to be adsorbed across adjacent Mn atoms, suggesting a preference for a pathway that resembles concerted proton-electron transfer. Conversely, when surface protons are present, the formation of *OOH and subsequent O₂ is less favourable, resulting in a higher overpotential.

Furthermore, we have incorporated the scaling relationships between *OH and *OOH adsorption energies into our analysis. These relationships are critical for understanding the overpotential required for the OER and are indicative of the change in the reaction mechanism. By plotting the various free energy changes for different proton configurations (Figure R5), we observe that the absence of protons leads to a lower overpotential and potentially facilitates a concerted mechanism.

The insights from these calculations contribute to a comprehensive theoretical framework that supports the experimental observations of an electric field-facilitated change in the OER mechanism on α -MnO₂ nanowires. While our DFT calculations do not directly simulate a PCET process, they do provide valuable information on the surface states and energy barriers that are consistent with such a mechanism.

We have amended our manuscript to include these detailed computational insights, ensuring a coherent narrative that bridges the experimental and theoretical aspects of our work. The revised sections, which can be found on pages 14 and 15 of the Supplementary Information, have been highlighted for ease of reference.

Figure R5. (*Supplementary Figure 14*) The relationship between the free energy of adsorption states on MnO₂ (010) facet and the different protons adsorption states and configuration of surface and tunnel protons. (a-d) show the colour map of ΔG_1 , ΔG_2 , ΔG_3 , ΔG_4 .

Figure R6. (*Supplementary Fig. 13*) The structure model of *OOH intermediates of different surface

and tunnel structures. Blue, red, and pink atoms represent Mn, oxygen, and proton, respectively. The adsorbed OOH is marked by a green colour. a, b, c and d illustrate the adsorption states with no surface protons, surface protons on site 1, surface protons on site 2 and surface protons on site 3, respectively. The numbers represent the number of protons in the tunnel structure.

*Revised main text (pages 14 and 15): “The deprotonation process is known to be important for OER, and if the deprotonation of *OH or *OOH is limited, it will directly affect the **potential-determining step (PDS)** (see details in Supplementary Notes). We calculated and compared ΔG_1 , ΔG_2 , ΔG_3 , and ΔG_4 with different proton configurations (Supplementary Fig. 14). Considering various proton configurations on the surface and internal structures, we aimed to provide a comprehensive evaluation of how proton transfer influences oxygen evolution. ΔG_3 reveals an intriguing phenomenon in the adsorption structure of *OOH. In models without surface protons, *OOH is found to be absorbed by two adjacent Mn atoms (Supplementary Fig. 13), resulting in smaller ΔG_3 values compared to other configurations. This interesting finding also occurs in models with no tunnel protons but protons on opposite Mn sites. This suggests that the deprotonated surface favours the addition of OH^- to *OH. Conversely, if the oxygen on the target Mn sites is protonated, the formation of *OOH, as well as subsequent oxygen molecule formation, is challenging, leading to an overpotential of ~ 0.8 V (Supplementary Fig. 15). Additionally, tunnel protons can adjust the overpotential by affecting the adsorption free energy, although they cannot modify adsorbates (Supplementary Fig. 16). In conclusion, we found that deprotonated surface states are crucial for forming dual-site O-O and achieving a moderate theoretical overpotential. If considering the intermediates on $\alpha\text{-MnO}_2$ surface as a motif in the whole structure, the deprotonation process is thus determined by both the **redox properties** of $\alpha\text{-MnO}_2$ surface and H^+/OH^- concentration.”*

2. Experiments suggest that the gate electric field generates a high concentration of proton acceptors, OH^- . If I understood correctly, this would suggest that the deprotonated structure (MnO_2) is predominant since there are more proton acceptors. But the computed overpotential of (MnO_2) is larger than that of protonated (MnO_2+4H).

Response: We appreciate the reviewer's astute observation concerning the relationship between proton acceptors and the overpotential associated with different protonation states of MnO_2 . Our DFT calculations, indeed, initially suggested a higher overpotential for deprotonated MnO_2 compared to the fully protonated form (MnO_2+4H). This result appears to contrast with the experimental indication that an applied gate electric field enhances the concentration of OH^- ions, favouring a deprotonated catalyst surface.

Upon reevaluating our computational data with a more comprehensive view of surface proton configurations, we find that the situation is more nuanced. Specifically, the critical overpotential-determining steps are the third and fourth steps of the OER mechanism, which involve the formation of the O-O bond. This bond formation is facilitated on a deprotonated surface, analogous to the mechanism observed in Photosystem II (PSII), where the O-O bond forms between two manganese sites.

Further analysis shows that the presence of surface protons significantly affects the adsorption of the *OOH intermediate, leading to a higher Gibbs free energy change and, consequently, a higher overpotential. It follows that a surface with adjacent protons actually discourages the OER, contrasting with the deprotonated surface which promotes it.

Therefore, while the initial DFT calculations highlighted the overpotential for a fully deprotonated MnO_2 surface, a more detailed examination reveals that the actual electrocatalytic process is heavily

influenced by the specific surface proton configuration. Our revised calculations now provide a clearer and more accurate picture that aligns with the experimental observations of enhanced OER activity under the influence of an external electric field.

These insights have prompted us to revise the relevant sections of our manuscript, particularly on page 15, where we have delineated these findings in red for better visibility.

*Revised main text (page 15): “This suggests that the deprotonated surface favours the addition of OH⁻ to *OH. Conversely, if the oxygen on the target Mn sites is protonated, the formation of *OOH, as well as subsequent oxygen molecule formation, is challenging, leading to an overpotential of ~0.8 V (Supplementary Fig. 15). Additionally, tunnel protons can adjust the overpotential by affecting the adsorption free energy, although they cannot modify adsorbates (Supplementary Fig. 16). In conclusion, we found that deprotonated surface states are crucial for forming dual-site O-O and achieving a moderate theoretical overpotential.”*

3. The lattice oxygen mechanism has not been considered.

Response: We are grateful for the reviewer's recommendation to explore the lattice oxygen mechanism (LOM) in manganese oxides. The LOM is indeed recognised for its potential to explain the high activity of perovskite oxides in OER processes. This mechanism typically requires the oxygen 2p band to be positioned above the lower-Hubbard band (Figure R7a), which facilitates the release of oxygen from the lattice, as suggested by recent studies (*Nat. Chem.*, 2017, 9, 457–465; *Nat. Energy*, 2019, 4, 329–338).

In our study, we observed no abnormal positioning of the oxygen 2p band that would suggest a LOM pathway (Figure R7b). As such, our analysis primarily centred on the adsorbate oxygen evolution mechanism (AEM). However, we did not dismiss the possibility of direct O-O coupling, as evidenced by our DFT calculations indicating a free energy change corresponding to a 0.8 V overpotential (Figure R7c)—though this overpotential is too high to support direct O-O coupling as a dominant mechanism in our system.

Interestingly, our DFT calculations revealed a dual-site adsorption state for *OOH, diverging from traditional AEM and hinting at a Langmuir–Hinshelwood (LH) type mechanism. While superficially similar to direct O-O coupling, we postulate an acid-base nucleophilic attack mechanism instead. We also note the structural similarity between MnO₂ and the core complex of Photosystem II (PSII), which may imply analogous adsorbate behaviour.

To more thoroughly address the possibility of LOM and its implications, we have expanded our discussion in the manuscript. This includes a nuanced analysis of the dual-site *OOH adsorption state and its potential mechanistic pathways. These additions and revisions have been incorporated into the main text on page 14 and Supplementary Fig. 11, now highlighted in red.

Figure R7. (a) The schematic diagram of energy bands of Mott–Hubbard splitting and the O₂ release mechanism. UHB and LDH represent upper-Hubbard band (UHB) and lower-Hubbard band (LHB), respectively. (b) The calculated projected density of states (DOS) of pristine α -MnO₂. (c) Structure model of direct O–O coupling on MnO₂ surface.

*Revised main text (page 14): “As the analogue of the oxygen-evolving complex in Photosystem II, the OER pathway on α -MnO₂ is proposed as an adsorbate evolution mechanism, including three critical intermediates *OH, *O, *OOH. The lattice oxygen mechanism was ruled out due to no feature of rise O 2p band and the high free energy change of direct coupling of two oxygen sites⁴⁷ (Supplementary Fig. 11). It is worth noting that we found the dual sites absorption state of *OOH (Supplementary Figs. 12,13), attributed to Langmuir–Hinshelwood (LH) mechanism⁴⁸. LH mechanism on MnO₂ resembles Mn₄CaO_x in PSII, demonstrating a structural similarity induced by similar adsorbates.”*

New reference:

47. Huang, Z.-F. et al. Chemical and structural origin of lattice oxygen oxidation in Co–Zn oxyhydroxide oxygen evolution electrocatalysts. Nat. Energy 4, 329-338 (2019).

48. Wang, Z., Goddard, W. A. & Xiao, H. Potential-dependent transition of reaction mechanisms for oxygen evolution on layered double hydroxides. Nat. Commun. 14, 4228 (2023).

Revised Supplementary Fig. 11

Supplementary Figure 11. (a) The calculated projected density of states (DOS) of pristine α -MnO₂. (b) Structure model of direct O–O coupling on MnO₂ surface.

4. Raw structures must be reported for reproducibility and visualization. It is very difficult to evaluate any computational work just by looking at numbers in profiles (Figure S9). The convoluted description of bonding (page S6) would be easier to understand if structures/schemes were provided.

Response: Thanks for the suggestions, the raw structure models are presented in the revised manuscript.

The structure models can be found in Supplementary Figs. 12,13, marked by red colour.

Supplementary Figure 12. The structure models of intermediates in OER pathways on MnO_2 (010) facet with no surface proton. The first digit in labels (0, 2, 4) represents the structures with 0, 2 and 4 protons adsorbed on the di- μ_2 -oxo-O sites in a tunnel structure, respectively. The second digit in labels (1, 2, 3, 4) represents the pristine (010) facet, *OH, *O and *OOH, respectively.

Supplementary Figure 13. The lateral view of *OOH intermediates of different surface and tunnel structures. a, b, c and d illustrate the adsorption states with no surface protons, surface protons on site 1, surface protons on site 2 and surface protons on site 3, respectively.

Other technical issues raised by Reviewer #2:

1. I assume they performed spin-polarized calculations, but this is not indicated in the text. Also, magnetic moments for relevant Mn and O atoms should be reported.

Response: Thanks for the valuable suggestions, we have added the description to the revised experimental methods. The magnetic moments of Mn and O atoms are also added, as shown in Figure R8.

The magnetic moments can be found in Supplementary Fig. 8, marked by red colour.

Figure R8. The structure model of MnO₂ without and with surface protons. The labels are magnetic moment (μ_B). Blue, red, and pink spheres represent Mn, O and H atoms, respectively.

2. *It would be useful to cite previous DFT+U work to support the choice of $U = 4.5$ V for this material.*

Response: The reference is *Angewandte Chemie*, 2013, 125, 2534-2537.

The reference has been added to Supplementary Information, No.8 reference marked in red colour.

3. *Please report details about the numerical computation of vibrational frequencies.*

Response: The Raman shift of MnO₂ is mostly relied on experimental results. Here, we present the analysis of the Raman modes in a typical α -MnO₂. The Raman modes of α -MnO₂ contribute from Mn-O vibrations and 4 Mn and 8 O atoms are lying on an 8h site. Hence, the optical modes can be calculated as, $\Gamma_{\text{Mn-O}} = 6A_g + 6B_g + 3E_g + 2A_u + 3B_u + 5E_u$ (Table R1). The Raman active modes are A_g , B_g , and E_g . In our experiment, A_g modes can be observed and identified (~ 579 and ~ 632 cm^{-1}).

The related description has been added to Supplementary Information on page S4, marked in red colour.

Table R1: The chart of vibrations in a typical α -MnO₂ unit cell. Adapted from reference, *J. Phys. Chem. C*, 2008, 112, 13134–13140.

Site group C _s	correlation	Factor group C _{4h}	Activity
24 A'		6 A _g	$\alpha_{xx} + \alpha_{yy}, \alpha_{zz}$
		6 B _g	$\alpha_{xx} - \alpha_{yy}, \alpha_{zz}$
		3 E _g	(α_{xx}, α_{yy})
12 A''		3 A _u	T _z
		3 B _u	-
		6 E _u	(T _x , T _y)

Revised supplementary information (page S4): Raman spectra for the obtained sample are shown in Supplementary Fig. 1b. *The Raman modes of α -MnO₂ contribute from Mn-O vibrations and 4 Mn and 8 O atoms are lying on an 8h site. Hence, the optical modes can be calculated as, $\Gamma_{Mn-O} = 6A_g + 6B_g + 3E_g + 2A_u + 3B_u + 5E_u$. According to factor group analysis, the A_g, B_g and E_g modes are Raman active¹².*

4. Figure S9a: When they say “proton”, I assume they mean “hydrogen” since charges cannot be described in PBC. Then, when adding/removing H, what are the oxidation states of Mn atoms in each configuration? How can they affect the overpotential or correlate with the potential determining steps?

Response: We agree with the reviewer that the proton is not correct. We revised the description to "tunnel-adsorbed protons" in the legend of Supplementary Fig. 9.

For the second part, when adding hydrogen to Mn, the oxidation state of Mn is reduced (~+3), which can be deduced by the increased magnetic moment on Mn atoms (Figure R8). Regarding the effect of adding or removing protons on potential determining steps, we found that in most cases, PDS are steps 3 and 4 and they didn't show a strong relationship with protons configuration (Supplementary Fig.16). The effect of protons on overpotential is dominated by adjusting the adsorption state of *OOH. We found that the deprotonated surface states are essential to form dual sites O-O and achieve a moderate theoretical overpotential. On this basis, tunnel-adsorbed protons can further decrease the adsorption energy. Details can also be found in response to Comment 1.

5. Figure S9b: It is very difficult to discern the tonalities of blue. Please change the color palette or include the numerical values in the graph.

Response: Thanks for the suggestion, we have changed the colour palette of new Supplementary Fig. 15b to make it clear. Please see Figure R9 for the comparison.

Figure R9. The comparison of gradient blue palette (a) and revised colour palette (b).

6. *Nomenclature: It is “potential determining step” rather than “rate determining step”. Also, sometimes they refer to “barriers” but transition states are not computed, so this is not entirely correct.*

Response: We agree with the reviewer's comments, it should be "potential determining step (PDS)". The related content has been modified. To make it entirely correct, we modified the related description about "barriers", it should be "free energy change".

7. *Figure S10a: typos in X axis, it should be “site” instead of “sitie”.*

Response: Thank you for pointing out this typo, the errors have been corrected in the new Supplementary Fig. 16.

8. *Figure S11c: They present a regression line with only three points.*

Response: We greatly appreciate the concern about Supplementary Fig. 11 from the reviewer. Upon revisiting our experimental data, we recognised that the initial results may have suffered from a suboptimal linear relationship, possibly due to the decreased ionic strength of electrolyte. To address this, we conducted further electrochemical measurements in KOH electrolyte across various pH levels, using potassium sulphate as a supporting electrolyte to maintain a consistent concentration of K^+ ions.

The updated pH-dependent electrochemical analysis is presented in Figure R10. We recalculated the reaction order (ρ) using the following expression:

$$\rho = \left(\frac{\partial \log j}{\partial \text{pH}} \right)_E = - \left(\frac{\partial E}{\partial \text{pH}} \right)_i / \left(\frac{\partial E}{\partial \log j} \right)_{\text{pH}}$$

The new data reveals a clear first-order reaction with a fitted slope of 1.03 ± 0.06 . This suggests a reverse first-order dependence on H^+ , indicating that the Oxygen Evolution Reaction (OER) is significantly influenced by the concentration of the H^+ acceptor, namely OH^- .

In light of these new findings, we conclude that the OER process on the MnO_2 surface exhibits an inverse first-order dependence on H^+ concentration. The manuscript has been updated accordingly to reflect these findings on pages 15 and 16, and the revised data in Supplementary Fig. 17, with changes highlighted in red for clarity.

Figure R10. (a) pH dependence for CV curves of MnO₂ in KOH solution with different concentrations. (b) The relationship between current density at 1.8 V vs. RHE and pH.

Revised main text (pages 15 and 16): “To understand the PCET process of OER, the OER activity of MnO₂ at different pH was measured (Supplementary Fig. 17a,b). Reaction order (ρ) can be determined by the linear relationship between $\log j$ and pH, $\rho = \left(\frac{\partial \log j}{\partial \text{pH}} \right)_E = - \left(\frac{\partial E}{\partial \text{pH}} \right)_i / \left(\frac{\partial E}{\partial \log j} \right)_{\text{pH}}$, where j is current density, E is potential versus RHE. The fitted slope value is 1.03 ± 0.06 , presenting the reverse first-order dependence on H^+ . Hence, the rate-determine step of $\alpha\text{-MnO}_2$ for OER is decided by the concentration of H^+ involved in reaction, resulting in a strong pH dependence OER activity (i.e., decoupled proton-electron transfer). CV curves at different pH also gives some information about redox transition of Mn (Supplementary Fig. 17c,d). The separation of oxidation and reduction peak potential shows a dependence on pH and scan rate. With the increase of pH, the width of CV peaks and the redox potential separation decreases at the same scan rate, demonstrating a decreased polarisation of proton-electron reaction at resting state before OER. The results above demonstrate the OER process on the MnO₂ surface is an uncoupled proton-electron transfer reaction, which shows inverse first-order dependence on H^+ concentration. This also indicates that increasing the H^+ acceptor can adjust the electron and proton coupling to enhance OER. Herein, the energy profile of the OER process is dominated by the proton configuration, and maintaining a circulation of lattice and surface protons will contribute to moderate adsorption energy OER thermodynamics.”

Revised Supplementary Figure 17:

Supplementary Figure 17. pH-dependent OER measurement. (a) pH dependence for CV curves of MnO₂ in KOH solution with different concentrations. The electrolyte was prepared by adding potassium sulphate to maintain the constant K⁺ strength. (b) The relationship between current density at 1.8 V vs. RHE and pH. (c) The CV curves of MnO₂ were measured at different pH KOH solutions with different scanning rates. (d) The corresponding peak separation at different scan rates. E_{peak}-E_{eq} represents the difference between peak potential and equilibrium potential.

Response to Reviewer #3

This study reports a concerted proton-electron transfer strategy for enhancing the OER activity of α -MnO₂ through the apply of an external electric field during the reaction process. This work is well done with a coherent logic flow, the in situ study is conducted in detail, and the key finding is interesting. However, it lacks sufficient novelty as the protonation/deprotonation processes of the prototype α -MnO₂ material have been well-established in Zn-MnO₂ batteries, and the conclusions need to be strengthened by more rational experimental and characterization designs. In addition, although the water electrolysis employing the concept developed in this study has been successfully demonstrated, its practical competitiveness remains an issue. Before its publication, the following concerns should be well addressed.

Response: We thank the Reviewer for acknowledging the thoroughness of our study and the interesting nature of our findings. We also appreciate the Reviewer's insights and would like to share additional thoughts regarding the novelty and application of our work.

Our research introduces a concept of utilising an external electric field to influence proton dynamics during the OER process. This approach, inspired by nature's own efficiency in catalytic processes, is indeed an extension of the principles observed in Zn-MnO₂ batteries. However, the dynamic control we propose offers a fresh perspective by actively and reversibly influencing the reaction environment, which is a new exploration in the realm of electrocatalysis.

The rich and intricate nature of proton involvement in various electrochemical processes indeed calls for a deeper understanding. Our work takes a step in this direction by trying to disentangle the intertwined proton-related mechanisms and their roles in electrocatalysis, which, as the reviewer pointed out, is an important yet challenging task.

We are excited about the prospects of this approach and are motivated to continue refining the technology. We acknowledge that while our current work demonstrates the foundational concept, there is a path ahead to translate these findings into competitive practical applications.

We have incorporated these additional considerations into our manuscript to better emphasise the innovative aspects and the potential impact of our work. These updates can be found highlighted on pages 3 and 4.

Revised main text (pages 3 and 4): "Research into proton insertion/extraction within the MnO₂ lattice has been ongoing for many years, particularly since the advent of alkaline batteries³³. However, in this new system, a more profound comprehension of the intricate and diverse properties of protons involved in various electrochemical systems, as well as the proton-electron processes, is imperative to attain enhanced control over the thermodynamics of proton reactions."

1. I am wondering whether the in situ Raman spectra can be accurately recorded at specific voltages using the CV technique, why not using the chronoamperometry technique at a certain constant potential?

Response: We appreciate the Reviewer's thoughtful suggestion regarding the use of chronoamperometry for *in situ* Raman measurements. Our decision to employ CV with a low scanning rate of 0.0005 V/s was made with careful consideration of the experimental objectives and constraints.

The primary advantage of our chosen method is its ability to capture a wide range of potential states, particularly transitional states that might be missed in a constant potential setup. By adjusting the acquisition and accumulation times, we ensured that each spectrum was measured over a 50 mV potential range within approximately 100 seconds. This approach provided sufficient resolution to

discern the sequential reaction processes and observe any structural evolution associated with the electrochemical reactions.

It's important to note that electrochemical reactions are potential-dependent, with reaction rates generally increasing with higher overpotentials, as described by the Butler-Volmer equation. Our CV approach allows us to explore these dynamics across a spectrum of potentials, capturing the structural changes that occur as the potential exceeds the equilibrium point and continues to rise.

While we acknowledge that chronoamperometry offers the advantage of precise measurements at a fixed potential, which is ideal for identifying specific reaction intermediates, it tends to overlook the transient states that occur between established potential points. Given the current time resolution limitations of our Raman setup, CV provides a more comprehensive overview of the reaction landscape, including both stable and transitional states.

We recognise that if our Raman measurement's time resolution could be enhanced to less than a second, chronoamperometry would indeed be an invaluable method for pinpointing precise intermediates. This consideration will certainly inform our future experimental designs and methodological choices.

The manuscript has been updated to include a more detailed explanation of our choice of CV for *in situ* Raman spectroscopy, highlighting the advantages and limitations of this approach. These updates can be found on page 7, marked in red for easy reference.

Revised main text (page 7): “To investigate the structural evolution, we recorded the Raman spectra during the cyclic voltammetry (CV) measurement with a potential range from 1.0 to 1.7 V vs. RHE (Fig. 1d and Supplementary Fig. 4). By adjusting the CV scanning rate, we ensured that each spectrum was measured over a 50 mV potential range. This approach provided sufficient resolution to discern the sequential reaction processes and observe any structural evolution associated with the electrochemical reactions.”

2. If there is bubble interference at 1.7 V, then the Raman spectrum recorded at 1.7 V can be excluded from Fig. 1d, as it is not relevant to the subsequent discussion.

Response: Thank you for your suggestion regarding the Raman spectrum at 1.7 V. We understand your concern about the potential interference from oxygen bubbles at this voltage. However, we believe it is important to include this data point in our analysis.

The reduced intensity of the Raman spectrum at 1.7 V is indeed a result of oxygen bubble generation, which is an inherent part of the oxygen evolution reaction under these conditions. We have chosen to retain this spectrum as it provides valuable context for understanding the cathodic process during the voltage sweep from 1.7 V to 1.0 V. This aspect of our experiment is particularly relevant to Comment 4, and we have provided a detailed explanation in our response to that comment.

Including the 1.7 V data point, despite its reduced intensity, offers a more comprehensive view of the electrochemical and structural changes occurring throughout the entire voltage range studied. We believe that this inclusion enhances the reliability and completeness of our analysis.

We have updated our manuscript to clearly explain the reasoning behind retaining the 1.7 V spectrum in Fig. 1d, ensuring that readers understand its relevance to the overall study. These clarifications can be found in the section related to Fig. 1d, now highlighted for easy reference.

Revised main text (pages 7 and 8): “When the potential reaches ~1.6 V, we notice that large oxygen bubbles are generated from the electrode surface, scattering the laser and causing a weak or

inaccurate Raman signal indicating a typical oxygen evolution reaction feature. While the spectrum intensity is low, it provides evidence of the presence of oxygen bubbles and offers insight into the structural characteristics.”

3. What's the reason for the insignificant changes of the peaks at the first cycle from 1.0-1.7 V compared with those in the second cycle within the same potential window as shown in Fig. 1d?

Response: Thank you for your keen insight into this phenomenon. According to our measured results of hydrogen concentration, the protons concentration in pristine α -MnO₂ is not high and shows no obvious indication of protonated manganese oxides feature. In this case, the deprotonation process in the first anodic process is not easy to observe. After the anodic process in the first cycle, the MnO₂ at low potential is highly protonated. This is the reason why we observed a significant variation in the second cycle. We added the explanation to the description related to ERD results, marked in red for easy reference.

Revised main text (page 12): “This suggests that the pristine MnO₂, despite being synthesized in an aqueous environment, is not heavily protonated. This observation explains why the first anodic process during in situ Raman measurement did not exhibit significant structural evolution (Fig. 1d).”

4: In the illustration of the in situ Raman spectra, the authors claim that “after the anodic process, the profiles return to the typical doublet vibration bands”. However, this conclusion is not very persuasive, as at the initial stage of the cathodic process, this peak is even invisible at 1.6 V.

Response: Thank you for pointing out the need for clarification regarding the changes observed in the Raman spectra in Fig. 1d. We acknowledge that the description in our initial manuscript might have been unclear, and we appreciate the opportunity to provide a more detailed explanation.

In our study, the cyclic voltammetry process was divided into three distinct parts for analysis:

Cathodic Process (1.6 V to 1.2 V): During this phase, the Raman signal at the higher potential of 1.6 V was affected by oxygen bubble formation. This led to a scattering of the Raman signal and a resultant weak spectrum.

Redox Process (1.2 V to 1.0 V to 1.4 V): This part involved a redox transition, where the Raman spectra displayed certain characteristic changes.

Anodic Process (1.4 V to 1.6 V): It was during this stage in the second cycle that we observed the Raman bands recovering to their doublet state.

The recovery of the Raman spectra to doublet bands, which we refer to in our conclusion, specifically occurs in the anodic process of the second cycle. The initial lack of significant changes in the first cycle can be attributed to the interfering effect of the oxygen bubbles generated at higher potentials, which obscure the Raman spectra. By the second cycle, these effects are mitigated, allowing for clearer observation of the spectral changes.

We have revised our manuscript to articulate this explanation more clearly, especially in the sections discussing the cyclic voltammetry process and the interpretation of the Raman spectra.

Revised main text (page 7): “Two cycles were measured to provide a comprehensive understanding of structural evolution, taking into account the initial states before electrochemical conditioning, thereby ensuring robust and convincing results.”

Revised main text (page 8): “Following the second anodic process (process 3), the profiles, including peak position and intensity, revert to the typical doublet vibration bands characteristic of α -MnO₂.”

5. Could the author explain the different peak shifts of the ν_2 peak within the potential range of 1.2-1.0-1.2 V as shown in Fig. 1f? In my opinion, it is not a reversible fluctuation as claimed in the manuscript.

Response: Thank you for highlighting the need for clarity regarding the shifts of the ν_2 peak within the potential range of 1.2-1.0-1.2 V. Upon re-examination, we agree that our initial interpretation of these shifts as reversible fluctuations may have been overly simplistic.

The observed shift to a higher wavenumber and the increase in intensity of the ν_2 band indicate a transition from an octahedral structure to a tetragonal one, resembling the spinel structure of Mn₃O₄. This change is attributed to the Jahn–Teller effect of Mn³⁺ ions, which induces a distortion from octahedral to tetrahedral coordination at lower potentials (1.2 - 1.0 V). While the Raman peak shape suggests some degree of structural recovery during the anodic process, a careful comparison of the peak positions reveals subtle differences.

Therefore, we acknowledge that the reversibility of the ν_2 peak, and by extension, the associated structural evolution, cannot be definitively demonstrated. In our discussion of proton-related structural evolution (Fig. 2), we focus on the notion that proton cycling may not be highly reversible. This insight aligns with our findings regarding the ν_2 peak shifts.

To accurately reflect these observations and our current understanding, we have updated our manuscript by removing references to the "reversibility" of these changes. We believe this revision provides a more precise and nuanced interpretation of the data.

These changes have been made in the relevant sections of the manuscript, specifically in the discussion related to Fig. 1f, and are now highlighted for clarity.

Revised main text (page 8): “It is noteworthy that the ν_2 peaks exhibit low reversibility in terms of wavenumber, attributed to the irreversible structure transformation induced by potential Jahn–Teller distortion. Following the second anodic process (process 3), the profiles, including peak position and intensity, revert to the typical doublet vibration bands characteristic of α -MnO₂.”

6. To better support the structural evolution, I suggest to conduct the in situ XAS study in a similar potential trip as that for in situ Raman spectra.

Response: Thank you for your valuable suggestion to conduct *in situ* XAS studies using a similar potential trip as our *in situ* Raman experiments. We understand the potential benefits of such an approach for corroborating our findings on structural evolution.

As we discussed earlier in response to a different comment, our Raman measurements were designed to capture transient states during continuous potential scanning, revealing subtle transition states. However, the implementation of a similar methodology in XAS presents significant challenges. The acquisition time for a single spectrum in XAS, particularly when obtaining acceptable quality XANES (X-ray Absorption Near Edge Structure) and EXAFS (Extended X-ray Absorption Fine Structure) data in fluorescence mode, typically exceeds 12 minutes. This time frame makes it difficult to perform XAS measurements during dynamic potential scanning, as we did with Raman spectroscopy.

To adapt to these constraints, we instead conducted *in situ* XAS at three select potential steps, which allowed us to compare the structural states of the material post-deprotonation. These potential steps

were chosen to provide insights into key stages of the electrochemical process. Additionally, we carried out *in situ* XAS experiments using multi-step chronoamperometry at 0.9 V and 1.6 V vs. RHE to further investigate the structural changes.

We believe these *in situ* XAS measurements, although conducted at fixed potential points rather than during continuous scanning, effectively support our conclusions. They provide crucial insights into the structural evolution of the material under different electrochemical conditions, complementing our Raman spectroscopy findings.

We have revised our manuscript to include a more detailed explanation of the methodological considerations and choices for our *in situ* XAS experiments.

Revised main text (page 9): To further analyse the structural changes in the coordination environment of Mn atoms, in situ X-ray absorption spectroscopy (XAS) (Supplementary Fig. 5) was performed during the anodic process at three potentials (0.9, 1.3, and 1.6 V). In order to attain high-quality spectra, time-resolution was not prioritised in the in situ XAS measurements. Instead, multi-potential measurements were employed.

7. In Fig. 1g, the Mn-O bond length exhibits a slight increase from 0.9 to 1.3 V, rather than showing a decrease as stated. What's more, what's the reason for the enhanced scattering intensity of the Mn-O bond with the increasing potential?

Response: Thank you for highlighting the Mn-O bond trend. From 0.9 to 1.3 V, the peak maximum position of Mn-O indeed exhibits a slight increase, followed by an apparent decrease at 1.6 V. We hypothesise that during deprotonation, certain Mn sites undergo a relaxation process to restore octahedral structures. Consequently, some low-valent Mn sites (+3) may retain Jahn-Teller octahedral distortion. The elongation distortion of the MnO₆ octahedron has also been observed in the atomic structure, with the z ligand bonds longer than the four planar bonds. Hence, it is reasonable to expect a slight increase in the average Mn-O bond length in MnO₂.

Our DFT calculations also prove this point, the elongation of Mn-O bond length is demonstrated to be 2.43 Å whereas the other Mn-O bond lengths are ~1.90 Å. Please find more information in Response to Comment 8 and Figure R11.

The second point pertains to the scattering intensity of Mn-O. According to the theory of extended X-ray absorption fine structure (EXAFS) and Wavelet analysis, the observed increase in scattering intensity can be attributed to multiple scattering or multi-electronic excitations (*Am. Min.*, 2003, 88, 694-700; *Phys. Rev. B*, 2005, 71, 094110). However, the physical explanation of this phenomenon at high potential without the addition or substitution of foreign atoms is unclear. In this context, the recovery of Mn-Mn coordination is supposed to be achieved through the formation of bridging oxygen, thereby facilitating the restoration of Mn-Mn connections. Therefore, we believe these findings are consistent.

We have revised our manuscript to present precise descriptions of *in situ* XAS experiments.

Revised main text (pages 9 and 10): "Compare the spectrum at 0.9 and 1.6 V, Mn-O bonds and Mn-Mn coordination shift to low apparent radial distance, indicating a decrease in Mn-O length and the Mn-Mn distance. At 1.3 V, a slight increase in Mn-O distance is observed, which could be linked to the elongation of z ligand bonds resulting from Jahn-Teller octahedral distortion."

8. There is an issue regarding the transformation from a double octahedra configuration to a partial

tetrahedra configuration as depicted in Fig. 1h. When one of the protonated bridge O (di- μ -oxo-O) that shared by two octahedra detached from another, the octahedra remain intact (i.e. the coordination number of O, etc.).

Response: Thank you for bringing up the importance of discussing the structural transformation in more detail. We recognise the significance of elaborating on this issue. As indicated, the proposed transformation mechanism was inferred from *in situ* Raman results, where distinctive spinel structure features were observed. To substantiate this mechanism, we systematically constructed a series of protonated MnO₂ models to elucidate the role of protons.

As illustrated in Figure R11, when surface protons are adsorbed on sites 2 and 3, the [MnO₆] framework remains robust. Conversely, when surface protons are adsorbed on site 1, some di- μ -oxo-O bridges are disrupted, leading to the formation of low-coordination [MnO_x]. Notably, upon removal of terminal protons, the restoration of bridging oxygen is observed, accompanied by the formation of elongated Mn-O bonds, indicative of Jahn-Teller octahedral distortion. Additionally, our findings demonstrate that if terminal oxygen continues to react with OH⁻ and form *OOH, the di- μ -oxo-O bridges are also broken.

Based on these computed models, we provide compelling evidence for a structural transformation involving proton configuration evolution and a decrease in coordination number. While we acknowledge the possibility raised by the reviewer, we remain confident that our proposed mechanism aligns with both our experimental observations and computational results.

The related discussion has been incorporated into the main text on page 10, and Supplementary Figure 8, highlighted in red, has been added.

*Revised main text (page 10): “Therefore, in situ spectra results mainly demonstrate that the incorporated protons couple with the structure evolution edge-shared [MnO₆] octahedra accompanied by redox transition of Mn. We used Density functional theory (DFT) calculations to find the adsorption sites of protons and different models of proton adsorption on di- μ -oxo-O and mono- μ -oxo-O sites we built (Supplementary Figs. 6 and 7). We found that the protons adsorption on di- μ -oxo-O are thermodynamically spontaneous, while mono- μ -oxo-O sites is an energy unfavourable situation, indicating that di- μ -oxo-O acts as Brønsted basic sites. It is also interesting to find the break of bridging di- μ -oxo-O with protonated terminal oxygen sites, resulting low-coordinated corner-shared [MnO_x] polyhedrons (Supplementary Fig. 8). It can be recovered with the deprotonation of terminal oxygen and break again with *OOH formation by nucleophilic attack. Such results on proton adsorption sites and the related structural evolution effectively demonstrate the phenomenon observed by in situ spectroscopy characterisations.”*

Figure R11. The structure model of MnO_2 without and with surface protons on different sites. Blue, red, and pink spheres represent Mn, O and H atoms, respectively. e and f are the following steps d in the OER process. d represents the adsorption state of $^*\text{OH}$, e is deprotonation of $^*\text{OH}$ and f is the formation of $^*\text{OOH}$.

9. The possibility of this partial transformation from octahedra to tetrahedra contributing to a typical lattice-O involved OER mechanism should be considered. As an obvious pH-dependent OER activity is observed in this study, the authors are encouraged to elucidate the pH effect in depth.

Response: We are grateful for the reviewer's recommendation to explore the lattice oxygen mechanism (LOM) in manganese oxides. The LOM is indeed recognised for its potential to explain the high activity of perovskite oxides in OER processes. This mechanism typically requires the oxygen 2p band to be positioned above the lower-Hubbard band (Figure R12a), which facilitates the release of oxygen from the lattice, as suggested by recent studies (*Nat. Chem.*, 2017, 9, 457–465; *Nat. Energy*, 2019, 4, 329–338).

In our study, we observed no abnormal positioning of the oxygen 2p band that would suggest a LOM pathway (Figure R12b). As such, our analysis primarily centred on the adsorbate oxygen evolution mechanism (AEM). However, we did not dismiss the possibility of direct O-O coupling, as evidenced by our DFT calculations indicating a free energy change corresponding to a 0.8 V overpotential (Figure R12c)—though this overpotential is too high to support direct O-O coupling as a dominant mechanism in our system.

Interestingly, our DFT calculations revealed a dual-site adsorption state for $^*\text{OOH}$, diverging from traditional AEM and hinting at a Langmuir–Hinshelwood (LH) type mechanism. While superficially similar to direct O-O coupling, we postulate an acid-base nucleophilic attack mechanism instead. We also note the structural similarity between MnO_2 and the core complex of Photosystem II (PSII), which may imply analogous adsorbate behaviour.

To more thoroughly address the possibility of LOM and its implications, we have expanded our discussion in the manuscript. This includes a nuanced analysis of the dual-site $^*\text{OOH}$ adsorption state

and its potential mechanistic pathways. These additions and revisions have been incorporated into the main text on page 13 and Supplementary Figure 11, now highlighted in red.

*Revised main text (page 14): “As the analogue of the oxygen-evolving complex in Photosystem II, the OER pathway on α -MnO₂ is proposed as an adsorbate evolution mechanism, including three critical intermediates *OH, *O, *OOH. The lattice oxygen mechanism was ruled out due to no feature of rise O 2p band and the high free energy change of direct coupling of two oxygen sites⁴⁶ (Supplementary Fig. 11). It is worth noting that we found the dual sites absorption state of *OOH (Supplementary Figs. 12,13), attributed to Langmuir–Hinshelwood (LH) mechanism⁴⁷. LH mechanism on MnO₂ resembles Mn₄CaO_x in PSII, demonstrating a structural similarity induced by similar adsorbates.”*

New reference:

46. Huang, Z.-F. et al. Chemical and structural origin of lattice oxygen oxidation in Co–Zn oxyhydroxide oxygen evolution electrocatalysts. Nat. Energy 4, 329-338 (2019).

47. Wang, Z., Goddard, W. A. & Xiao, H. Potential-dependent transition of reaction mechanisms for oxygen evolution on layered double hydroxides. Nat. Commun. 14, 4228 (2023).

Figure R12. (a) The schematic diagram of energy bands of Mott–Hubbard splitting and the O₂ release mechanism. UHB and LDH represent upper-Hubbard band (UHB) and lower-Hubbard band (LHB), respectively. (b) The calculated projected density of states (DOS) of pristine α -MnO₂. (c) Structure model of direct O–O coupling on MnO₂ surface.

For the second part, we sincerely appreciate the insightful suggestions provided by the reviewer regarding the investigation of pH effects. Upon revisiting our experimental data, we recognised that the initial results may have suffered from a suboptimal linear relationship, possibly due to the decreased ionic strength of electrolyte. To address this, we conducted further electrochemical measurements in KOH electrolyte across various pH levels, using potassium sulphate as a supporting electrolyte to maintain a consistent concentration of K⁺ ions.

The updated pH-dependent electrochemical analysis is presented in Figure R13a,b. We recalculated the reaction order (ρ) using the following expression:

$$\rho = \left(\frac{\partial \log j}{\partial \text{pH}} \right)_E = - \left(\frac{\partial E}{\partial \text{pH}} \right)_i / \left(\frac{\partial E}{\partial \log j} \right)_{\text{pH}}$$

The new data reveals a clear first-order reaction with a fitted slope of 1.03 ± 0.06 . This suggests a reverse first-order dependence on H⁺, indicating that the Oxygen Evolution Reaction (OER) is significantly influenced by the concentration of the H⁺ acceptor, namely OH⁻.

We further conducted measurements of cyclic voltammetry (CV) curves at various pH levels (see Figure R13c). The observed redox peaks correspond to the transition between Mn^{3+} and Mn^{4+} . Notably, the separation between oxidation and reduction peak potentials exhibits dependence on both pH and scan rate (Figure R13d). In a high-pH solution (pH = 13.99), the difference ($E_{\text{peak}}-E_{\text{eq}}$) is smallest at the same scan rate, coinciding with the sharpest oxidation peaks. This suggests a reduced polarisation effect of protonation/deprotonation. These findings collectively indicate that the oxygen evolution reaction (OER) process on the MnO_2 surface involves a decoupled proton-electron transfer reaction, displaying an inverse first-order dependence on H^+ concentration. Moreover, the results suggest that increasing the H^+ acceptor can adjust the coupling between electrons and protons to enhance OER efficiency.

In light of these new findings, we conclude that the OER process on the MnO_2 surface exhibits an inverse first-order dependence on H^+ concentration. The manuscript has been updated accordingly to reflect these findings on pages 15 and 16, and the revised data in Supplementary Fig. 17, with changes highlighted in red for clarity.

Revised main text: (pages 15 and 16) “To understand the PCET process of OER, the OER activity of MnO_2 at different pH was measured (Supplementary Fig. 17a,b). Reaction order (ρ) can be determined by the linear relationship between $\log j$ and pH, $\rho = \left(\frac{\partial \log j}{\partial \text{pH}}\right)_E = -\left(\frac{\partial E}{\partial \text{pH}}\right)_i / \left(\frac{\partial E}{\partial \log j}\right)_{\text{pH}}$, where j is current density, E is potential versus RHE. The fitted slope value is 1.03 ± 0.06 , presenting the reverse first-order dependence on H^+ . Hence, the rate-determine step of $\alpha\text{-MnO}_2$ for OER is decided by the concentration of H^+ involved in reaction, resulting in a strong pH dependence OER activity (i.e., decoupled proton-electron transfer). CV curves at different pH also gives some information about redox transition of Mn (Supplementary Fig. 17c,d). The separation of oxidation and reduction peak potential shows a dependence on pH and scan rate. With the increase of pH, the width of CV peaks and the redox potential separation decreases at the same scan rate, demonstrating a decreased polarisation of proton-electron reaction at resting state before OER. The results above demonstrate the OER process on the MnO_2 surface is an uncoupled proton-electron transfer reaction, which shows inverse first-order dependence on H^+ concentration. This also indicates that increasing the H^+ acceptor can adjust the electron and proton coupling to enhance OER. Herein, the energy profile of the OER process is dominated by the proton configuration, and maintaining a circulation of lattice and surface protons will contribute to moderate adsorption energy OER thermodynamics.”

Figure R13. (Supplementary Figure 17) (a) pH dependence for CV curves of MnO₂ in KOH solution with different concentrations. The electrolyte was prepared by adding potassium sulphate to maintain the constant K⁺ strength. (b) The relationship between current density at 1.8 V vs. RHE and pH. (c) The CV curves of MnO₂ were measured at different pH KOH solutions with different scanning rates. (d) The corresponding peak separation at different scan rates. $E_{\text{peak}} - E_{\text{eq}}$ represents the difference between peak potential and equilibrium potential.

10. The role of the accommodated K⁺ needs to be clarified, and it is also necessary to investigate whether K⁺ cations in the electrolyte compete with protons for adsorption on the MnO₂ surface.

Response: We appreciate the reviewer's suggestion regarding the consideration of the role of K⁺. As outlined in the manuscript, the presence of accommodated K⁺ serves to stabilise the 2×2 tunnel structures and balance charges, thereby favouring the formation of mixed-valence states Mn³⁺/Mn⁴⁺.

Given the relatively low concentration of K⁺ (3% atomic percentage), direct interaction between K and Mn in affecting adsorbates can be disregarded. Similarly, insights into the role of K⁺ may be gleaned from the role of Ca in Mn₄CaO_x. Ca²⁺ is believed to be weakly bound to the Mn₄ cluster in PSII and involved in transition states by forming bridging oxygen between Mn–Mn and Mn–Ca. However, due to geometric limitations in the 2×2 tunnel structure, K⁺ is unlikely to form bridging oxygen between Mn–Mn. Nevertheless, in future investigations, we aim to replace K⁺ with other elements to explore if different tunnel-accommodated ions play a pivotal role in catalysis.

Regarding the competition between K⁺ and H⁺, we found no surface sites for the adsorption of K⁺ ions. Protons, on the other hand, can be adsorbed on terminal oxygen, and we calculated the free energy of this process, where H⁺ arises from the decomposition of water molecules. Our RBS data also corroborate a stable concentration of K⁺ within the MnO₂ structure (see Fig. 2 c-f), thereby validating proton incorporation. Another possibility is the potential involvement of K⁺ in the tunnel in the reaction. Literature indicates that the binding energy of K is 4 eV, whereas the binding energy of H₂O is 0.39 eV (*Chem. Phys. Lett.*, 2012, 544, 53–58), indicating that the incorporation of accommodated K⁺ is indeed challenging. In conclusion, the competition between K⁺ and H⁺ in our system is not readily apparent.

To make our statement correct, we added the revised description in page 12, highlighted in red colour.

Revised main text: (page 12) *“The consistent concentration of K^+ effectively rules out the possibility of K^+ intercalation or adsorption from the electrolyte.”*

11. What's the proton source for the protonation of MnO_2 in an alkaline medium of 1 M KOH? And as shown in Fig. 1, the protonation occurs at the cathodic process, while the OER runs at anodic process. Moreover, how to ensure its continuous supply during the OER process, i.e., maintaining a dynamic incorporation and deprotonation process in enhancing the OER?

Response: Thank you for addressing the concern regarding the source of protons. In alkaline electrolytes, water molecules serve as the source of protons, as detailed in the early literature on alkaline batteries (*J. Electrochemical Soc.* 112, 959, 1965). This reaction mechanism is illustrated schematically in Supplementary Fig. 23 (Figure R14a,b). Figure R14a illustrates the extraction process of protons from the lattice, where they hop on adjacent oxygen sites. Figure R14b depicts the adsorption process of H^+ , where H_2O dissociates into hydroxyl and adsorbed protons. These protons then couple with the oxygen in the lattice (μ -oxo-O). Furthermore, owing to the rotation and vibration of OH^- within the lattice, the O-H bonds are easily broken, allowing protons to jump to adjacent O sites or couple with hydroxyl. Furthermore, our DFT calculation results confirm that the spontaneous dissociation of water molecules leads to the formation of adsorbed protons, as depicted in Supplementary Fig. 6.

Regarding the second part of the comment, we have included a schematic illustration of the reaction mechanism of Mn_4CaO_x in PSII and MnO_2 in Figure R14c,d. The essential process in oxygen evolution involves the deprotonation of water molecules. On the surfaces of transition metal compounds, oxygen evolution reaction (OER) typically follows an inner sphere mechanism, where reactants form adsorbates through chemical bonding. In this context, surface catalysis resembles intrinsic electron-proton reactions. During the OER process, such as in step 1, protons are introduced to the surface by OH^- , highlighted in red to distinguish them from other oxygen atoms. Considering the dynamic nature of the surface and varying energy barriers for different steps, the adsorption of OH^- and the deprotonation of $*OH$ are not necessarily concerted. If $*OH$ remains on the surface, protons are retained. As the OER progresses, cyclic protonation and deprotonation processes sustain proton cycling.

In order to make the electron-proton mechanism clear, we added the revised description in supplementary information, highlighted in red colour.

Revised SI: (page S6): *“The discussion of the proton-electron mechanism of MnO_2 in alkaline media.*

Supplementary Fig. 23a illustrates the extraction process of protons from the lattice, where they hop on adjacent oxygen sites. Supplementary Fig. 23b depicts the adsorption process of H^+ , where H_2O dissociates into hydroxyl and adsorbed protons. These protons then couple with the oxygen in the lattice (μ -oxo-O). Furthermore, owing to the rotation and vibration of OH^- within the lattice, the O-H bonds are easily broken, allowing protons to jump to adjacent O sites or couple with hydroxyl.”

Figure R14. (a) The schematic presentation of the electron-proton mechanism. In the oxidation process, each Mn³⁺ ion loses an electron, and the protons will dissociate from oxygen and move to the adjacent oxygen. Ultimately, the protons couple with the hydroxyl and generate H₂O molecules. (b) In the reduction process, the protons are supported by the dissociated H₂O molecules at the solid/liquid interface and the protons are adsorbed by the bridged oxygen and introduced into the lattice. (c) The S states in the oxygen-evolution reaction. The oxygen-evolving complex is photo-oxidized through a series of S states to produce molecular oxygen from water. In the final steps before O=O bond formation, a new oxygen, O6, binds to the vacant site at Mn1. After a final photo-oxidation event, O5 and O6 appear poised to form an O=O bond, releasing molecular oxygen, reducing the cluster, and beginning the catalytic cycle anew. Glutamic acid at position 189 is noted as E189. This figure is adapted from *Science*, 2019, 366,305-306. (d) The reaction cycle of MnO₂ for OER reaction.

12. The economic issue of the overall water splitting using an external electric field with a high voltage remains. As it lacks evident advantage compared to other water electrolysis systems, particularly those capable of delivering ampere-level current density at low cell voltages.

Response: Thank you for highlighting this crucial point. According to our experimental findings, applying an external electric field effectively enhances the output power of overall water splitting. Our work primarily focuses on the application of external electric field enhancement strategies rather than investigating chemical strategies to improve the intrinsic activity of catalysts, while the output current density in our system may not be as high as in other catalyst systems.

However, it's worth mentioning that the external electric field enhancement strategies are not limited to manganese compounds alone. We have demonstrated their applicability in systems such as Ni(OH)₂, which exhibit higher current density, thus indicating their potential for widespread application. Given the capability to adjust proton-electron reactions, we believe that these strategies can be extended to various catalysis systems, including those with much higher current density.

In order to clarify this point, we added the revised description in the main text, highlighted in red

colour.

Revised main text: (page 22): *“Building upon this system, we have demonstrated the reproducibility of the proposed field-assisted water-splitting process, which holds promise for reducing the cost of commercial hydrogen production. Additionally, given the capacity to adjust proton-electron reactions within a flow reaction system, we believe this strategy can be applied to various industrial catalysis systems.”*

13. Several supplementary figures are not mentioned in the manuscript, such as supplementary Fig. 1 a-c.

Response: Thank you for checking the details. We have added the descriptions of Supplementary Fig. 1 in the main text.

Revised main text: (page 6): *“The typical structural feature of α -MnO₂ is the 2 × 2 tunnel structure stabilised by cations (Fig. 1b). Structure characterisations and element analysis of α -MnO₂ are presented in Supplementary Fig. 1a,b.”*

14. There exists a few typo errors or inappropriate expressions, the whole manuscript should be further polished.

Response: Thank you for the comment on technical issues. We have read through the manuscript and revised it carefully.

REVIEWERS' COMMENTS

Reviewer #1 (Remarks to the Author):

The manuscript has been revised according to the comments. Therefore, it can be accepted for publication.

Reviewer #2 (Remarks to the Author):

The authors have made a great effort to reply to my suggestions and comments and the computational part is now more coherent with respect to experiments. Still, two points need to be polished. After addressing those, I could recommend publication.

“4. Raw structures must be reported for reproducibility and visualization...”

The new Figure S12 is useful, but with “raw structures” I meant that the direct/Cartesian coordinates of all computed structures must be reported, either in the SI or in a digital repository. Readers must be able to openly access the raw data in order to reproduce the calculations.

“3. Please report details about the numerical computation of vibrational frequencies.”

I did not ask for the Raman modes, but for a detailed explanation of how the entropic terms (S_{vib} for solids and S_{vib} Strans Srot for molecules) of the free energies are computed.

Reviewer #3 (Remarks to the Author):

The authors have answered all my concerns and the quality of this manuscript has been greatly improved. I would like to recommend its publication as it is.

Response letter to the reviewers' comments

Manuscript ID: #NCOMMS-23-59958A

Response to Reviewer#1

The manuscript has been revised according to the comments. Therefore, it can be accepted for publication.

Response: We sincerely thank the reviewer for the positive feedback.

Response to Reviewer#2

The authors have made a great effort to reply to my suggestions and comments and the computational part is now more coherent with respect to experiments. Still, two points need to be polished. After addressing those, I could recommend publication.

Response: We thank the Reviewer for the recognition of revisions made to this manuscript. The detailed response for the remained comments are presented in the following response.

1. "4. Raw structures must be reported for reproducibility and visualization..."

The new Figure S12 is useful, but with "raw structures" I meant that the direct/Cartesian coordinates of all computed structures must be reported, either in the SI or in a digital repository. Readers must be able to openly access the raw data in order to reproduce the calculations.

Response: Thanks for the clarification about the "raw structures", we have attached the coordinates in the revised supplementary information.

Crystal structure data (atomic coordinates in Å).

_cell_length_a 5.7146

_cell_length_b 9.6150

_cell_length_c 24.0144

_cell_angle_alpha 90.0000

_cell_angle_beta 90.0000

_cell_angle_gamma 89.9943

Mn 0.25735 0.16882 0.25657

Mn 0.0072 0.66236 0.05114

Mn 0.25712 0.86084 0.11939

Mn 0.00729 0.35142 0.33305

Mn 0.25731 0.34001 0.04341

Mn 0.00764 0.84464 0.24709

Mn 0.25718 0.67412 0.32552

Mn 0.00736 0.1832 0.12712
Mn 0.75735 0.16863 0.25661
Mn 0.5072 0.66242 0.05114
Mn 0.75713 0.8609 0.11939
Mn 0.5073 0.35148 0.33305
Mn 0.75731 0.34007 0.04341
Mn 0.50662 0.8447 0.24709
Mn 0.75718 0.6741 0.32591
Mn 0.50736 0.18326 0.12712
O 0.25741 0.05464 0.12424
O 0.00597 0.55144 0.33924
O 0.25708 0.97057 0.25106
O 0.00724 0.46857 0.04628
O 0.25719 0.67755 0
O 0.00727 0.18426 0.20492
O 0.25731 0.33355 0.37938
O 0.00713 0.84565 0.17053
O 0.00746 0.15937 0.30948
O 0.2572 0.66355 0.10387
O 0.00711 0.85965 0.06666
O 0.25728 0.35926 0.27629
O 0.00736 0.20883 0.04722
O 0.25719 0.7075 0.24902
O 0.00681 0.80546 0.32812
O 0.25729 0.31438 0.12331
O 0.75741 0.0547 0.12424
O 0.50851 0.55151 0.33924
O 0.75708 0.97029 0.25122
O 0.50725 0.46863 0.04628
O 0.75719 0.67761 0
O 0.50743 0.18432 0.20492
O 0.75731 0.3325 0.37934
O 0.50714 0.84571 0.17053
O 0.50728 0.15943 0.30948

- O 0.75721 0.66361 0.10387
- O 0.50712 0.85971 0.06666
- O 0.75728 0.35938 0.27636
- O 0.50736 0.20889 0.04722
- O 0.75719 0.70731 0.2495
- O 0.50744 0.80552 0.32812
- O 0.75729 0.31444 0.12331

2. "3. Please report details about the numerical computation of vibrational frequencies."

I did not ask for the Raman modes, but for a detailed explanation of how the entropic terms (S_{vib} for solids and S_{vib} S_{trans} S_{rot} for molecules) of the free energies are computed.

Response: Thanks for the clarification about the numerical computation of vibrational frequencies.

We refer to Nørskov's method to calculate the Gibbs free energy. (*J. Phys. Chem. B* 2004, 108, 46, 17886–17892). For gas molecules, the entropy was taken from standard handbooks for gas-phase molecules. For adsorbed intermediates, the translation and rotation are hindered, and the entropy contribution is mainly from vibration. The entropy of surface adsorbed species is usually ignored in surface catalytic calculations. $S = R \left[\frac{\beta h c v}{e^{\beta h c v} - 1} - \ln(1 - e^{-\beta h c v}) \right]$, where R is ideal gas constant, $\beta = 1/k_B T$, h is Planck's constant, c is speed of light, v are vibration frequency. Considering a frequency of $\sim 500 \text{ cm}^{-1}$, S is $\sim 0.01 \text{ eV}$, which indicates the contribution of *OH, *O, and *OOH vibration frequency can be neglected in free energy calculation. We calculate the Gibbs free energy change (ΔG) of the reaction using the following equation: $\Delta G = \Delta E + \Delta ZPE - T\Delta S$, where ΔE , ΔZPE , T and ΔS are the difference in total energy difference between the reactant and the product, contributions of the zero-point energy to the free-energy change, temperature, and the change in entropy between the products and reactants, respectively. The contribution of entropy is very low, so previous work directly consider the zero energy without considering the contribution of entropy. We use frequency calculations to perform zero-point energy correction by $ZPE = 1/2 \times h v$, where h is Planck's constant and v is the frequency. We sum up the energy of all vibrations and calculate the zero-point energy using this equation. For the calculation of surface (*) and surface adsorbed species (*OH), we denote the ZPE of the surface as 0. When calculating the energy of adsorbed species, the atoms of surface are fixed and only the vibration of the adsorbed species is calculated. Hence, we calculated ΔG only calibrating ZPE of the contribution of adsorbed species. The zero-point energy corrections and entropic contributions to the free energies can be found in Table 1.

Table 1. Zero-point energy corrections and entropic contributions to the free energies. Entropies of gas-phase values are from *CRC Handbook of Chemistry and Physics, 49th ed.; Weast, R. C. Ed.; The Chemical Rubber Company: Cleveland, OH, 1968–1969; p D109*. The entropy of surface and adsorbed species is ignored. The ZPEs for the gas molecules the adsorbed species are taken from DFT calculations (frequency calculations).

	S (J K ⁻¹ mol ⁻¹)	ZPE (eV)
O ₂ (g)	205.147	0.095
H ₂ O(g) (0.035 bar)	216.819	0.56
H ₂ (g)	130.680	0.27
*	0	0
*OH	0	0.372
*OOH	0	0.445
*O	0	0.074
O+OH	0	0.450
*OH+*OH	0	0.754

Revisions in supplementary information: (Page S3) "*Zero-point energy correction was performed by frequency calculations, $ZPE = \frac{1}{2} \times hv$, where h is Planck's constant and v is the frequency. For the calculation of surface (*) and surface adsorbed species (*OH), we denote the ZPE of the surface as 0. When calculating the energy of adsorbed species, the atoms of surface are fixed and only the vibration of the adsorbed species is calculated. The zero-point energy corrections and entropic contributions to the free energies can be found in Table 1.*"

Response to Reviewer#3

The authors have answered all my concerns and the quality of this manuscript has been greatly improved. I would like to recommend its publication as it is.

Response: We sincerely thank the reviewer for the positive feedback.